# Remodelling of Rea1 linker domain drives the removal of assembly factors from pre-ribosomal particles

Johan Busselez [1,2,3,4], Geraldine Koenig[1,2,3,4], Carine Dominique[5],
Torben Klos [1,2,3,4], Deepika Velayudhan[1,2,3,4], Piotr Sosnowski [1,2,3,4,8],
Nils Marechal[1,2,3,4], Corinne Crucifix[1,2,3,4], Hugo Gizardin-Fredon [6,7],
Sarah Cianferani [6,7], Benjamin Albert [5], Yves Henry[5], Anthony K. Henras [5] &
Helgo Schmidt [1,2,3,4] ✉

The ribosome maturation factor Rea1 (or Midasin) catalyses the removal of assembly factors from large ribosomal subunit precursors and promotes their export from the nucleus to the cytosol. Rea1 consists of nearly 5000 amino-acid residues and belongs to the AAA+ protein family. It consists of a ring of six AAA+ domains from which the ≈1700 amino-acid residue linker emerges that is subdivided into stem, middle and top domains. A flexible and unstructured D/E rich region connects the linker top to a MIDAS (metal ion dependent adhesion site) domain, which is able to bind the assembly factor substrates. Despite its key importance for ribosome maturation, the mechanism driving assembly factor removal by Rea1 is still poorly understood. Here we demonstrate that the Rea1 linker is essential for assembly factor removal. It rotates and swings towards the AAA+ ring following a complex remodelling scheme involving nucleotide independent as well as nucleotide dependent steps. ATP-hydrolysis is required to engage the linker with the AAA+ ring and ultimately with the AAA + ring docked MIDAS domain. The interaction between the linker top and the MIDAS domain allows direct force transmission for assembly factor removal.

Ribosomes synthesise proteins and the production of ribosomes is the most energy consuming process in cells. In eukaryotes more than 200 assembly factors are involved in the production of ribosomes[1]. Ribosome assembly starts in the nucleolus with the transcription of the 5S and 35S rRNA precursors (pre-RNAs) by RNA polymerases III and I, respectively. The 35S pre-rRNA is subsequently cleaved into a smaller 20S and a larger 27S intermediate. The 20S precursor forms the basis of the future 40S ribosomal subunit, while the 27S precursor associates with various ribosomal proteins, ribosomal assembly and maturation factors as well as the 5S rRNA to form pre-60S ribosomal

(pre60S) particles[2]. These particles are subsequently exported to the cytosol via the nucleoplasm. During this process, they are gradually transformed into mature 60S subunits by transiently interacting with additional assembly factors that promote conformational rearrangements and rRNA processing steps[2]. The 60S assembly intermediates have been extensively studied using *Saccharomyces cerevisiae* as a model organism[3,4]. One of the earliest assembly intermediates identified in the nucleolus is a pre60S particle that carries the Ytm1 complex, consisting of Ymt1, Erb1 and Nop7[5] and involved in the processing of the 27S rRNA[6]. To promote transfer of this particle to the nucleoplasm,

[1]Institut de Génétique et de Biologie Moléculaire et Cellulaire, Integrated Structural Biology Department, Illkirch, France. [2]Centre National de la Recherche Scientifique, Illkirch, France. [3]Institut National de la Santé et de la Recherche Médicale, Illkirch, France. [4]Université de Strasbourg, Illkirch, France. [5]Molecular, Cellular and Developmental Biology Unit (MCD), Centre de Biologie Intégrative (CBI), CNRS, Université de Toulouse, Toulouse, France. [6]Laboratoire de Spectrométrie de Masse BioOrganique, IPHC UMR 7178, Université de Strasbourg, CNRS, Strasbourg, France. [7]Infrastructure Nationale de Protéomique ProFI – FR2048, Strasbourg, France. [8]Present address: BIOMEX, Siemensstrasse 38, 69123, Heidelberg, Germany. ✉e-mail: schmidth@igbmc.fr

the Ytm1 complex has to be removed, a reaction that is carried out by Rea1 (also known as "Ylr106p" or "Midasin")[7,8]. Rea1 associates with pre60S particles via its interaction with the Rix1 complex, which consists of Rix1, Ipi1 and Ipi3[9,10]. The Rea1 catalysed removal of the Ytm1 complex is essential for the nucleolar export of pre60S particles, because disrupting the interaction between Rea1 and Ytm1 leads to the accumulation of pre60S particles in the nucleolus[8].

In the nucleoplasm, pre60S particles bind to the assembly factor Rsa4, which tightly associates with another assembly factor, Nsa2. Nsa2 wraps around the H89 rRNA helix of the immature peptidyltransferase centre[11–13]. In addition to the Ytm1 complex, Rea1 also mechanically removes Rsa4 from pre60S particles[10]. In doing so, it might indirectly force Nsa2 to pull on the H89 rRNA helix to drag this important architectural feature of the peptidyltransferase centre into its correct position[11].

The Rea1 mediated Rsa4 removal is also crucial for the export of pre60S particles from the nucleoplasm to the cytosol. The removal of Rsa4 leads to an activation of the GTPase activity of Nug2[14]. Subsequently, Nug2-GDP dissociates, which allows the nuclear export factor Nmd3 to bind to the former Nug2 site. Nmd3 in turn associates with Crm1 and RanGTP to trigger the export of pre60S particles to the cytosol[14]. Disrupting the Rea1-Rsa4 interaction leads to the accumulation of pre60S particles in the nucleoplasm and severe growth defects in *S. cerevisiae*[10], demonstrating the importance of Rea1 also for nucleoplasmic pre60S particle export.

Rea1 is conserved from yeast to mammals and related to the motor protein dynein[15]. Its deletion in yeast leads to non-viable strains[15,16] underscoring the essential role of this complex molecular machine, which consists of nearly 5000 amino-acid residues. Rea1 folds into a hexameric ring of AAA+ (ATPases associated with various cellular activities) domains, each divided into large (AAAL) and small (AAAS) sub domains. The AAA+ domains AAA2-AAA5 contain all the conserved catalytic motifs for the hydrolysis of ATP[17,18]. AAA5 seems to be the main ATP-hydrolysis site in the AAA+ ring[18]. From the hexameric AAA+ ring a ≈1700 amino-acid linker domain emerges, consisting of stem-, middle- and top subdomains[17] (Fig. 1a, b). The linker top2 domain connects to a ≈600 amino-acid glutamate and aspartate rich region which ends in a ≈300 amino-acid metal ion dependent adhesion site (MIDAS) domain[15,17,19,20] (Fig. 1a, b). The MIDAS domain interacts with the ubiquitin like (UBL) domains of Ytm1 and Rsa4[8,10]. The interaction is mediated by a $Mg^{2+}$ ion and resembles the integrin MIDAS – ligand interaction[10,21]. The MIDAS domain is able to dock onto the AAA+ ring[17,20] and in the context of the Rea1-pre60S particle complex this docking site places the MIDAS domain in direct contact with the Rsa4 UBL domain[19]. Recently, the catch bond character of the MIDAS-UBL domain interaction has been demonstrated[22].

The question how force is transmitted to the substrate engaged MIDAS domain to remove Rsa4 or Ytm1 from pre60S particles has been controversial. Ytm1 and Rsa4 in-vitro release assays have established that the force production for assembly factor removal from pre60S particles requires ATP-hydrolysis[8,10], but the Rea1 conformations associated with ATP-hydrolysis have not been described yet. Early hypotheses suggested the Rea1 linker and MIDAS domain would move as a rigid body during ATP-hydrolysis to remove assembly factors from pre60S particles[10]. However, recent high-resolution cryoEM studies did not detect nucleotide dependent Rea1 linker remodelling[17,20]. It has been suggested that linker remodelling might not be relevant for assembly factor removal. Instead it was proposed that nucleotide driven conformational rearrangements in the AAA+ ring are directly communicated to the AAA+ ring docked, substrate engaged MIDAS domain to produce the force for assembly factor removal[20,23].

Here we provide evidence for the functional importance of the Rea1 linker region and demonstrate nucleotide independent as well as ATP-hydrolysis dependent steps of linker remodelling in Rea1. We show that linker remodelling consists of two main components, a rotation of the linker middle and top domains relative to the linker stem domain and a pivot movement towards the AAA+ ring. Furthermore, we demonstrate that linker remodelling is able to produce mechanical force. In the final remodelling step, the linker top interacts with the AAA+ ring docked MIDAS domain allowing direct transmission of force for assembly factor removal. Our results reveal key mechanistic events of one of the most complex eukaryotic ribosome maturation factors, whose mode of action has remained elusive since the first description of Rea1 20 years ago.

## Results
### The Rea1 linker top and middle domains are functionally important
To investigate the functional relevance of the Rea1 linker region, we created a Rea1 construct with deleted linker top and middle domains and fused the D/E rich region with the attached MIDAS domain directly to the linker stem (Rea1$\Delta_{middle-top}$) (Fig. 1c). We tested the functionality of Rea1$\Delta_{middle-top}$ using a modified version of a *Saccharomyces cerevisiae* GFP-Rpl25 pre60S export assay[10]. In this assay, the ability of a Rea1 construct to promote the nuclear export of pre60S particles is monitored by the detection of GFP fluorescence in the cytoplasm. We tagged the endogenous Rea1 gene with the auxin degron system, provided a plasmid encoding Rea1$\Delta_{middle-top}$ and monitored the cellular GFP fluorescence distribution after auxin induced degradation of endogenous Rea1. Consistent with the crucial functional role of the linker top and middle domains in pre60S particle nuclear export, we observed a drastic change in GFP fluorescence localization from the cytoplasm to the nucleus (Fig. 1d).

### The Rea1 linker undergoes nucleotide independent as well as nucleotide dependent remodelling
Having established the functional importance of the linker region, we decided to investigate the nucleotide dependent conformational changes of the Rea1 linker. Recently, we[17] and other groups[20] aimed at the determination of 3D structures of distinct linker conformations by electron microscopy (EM), but were not able to visualize linker remodelling. We reasoned that such an approach might fail to detect alternative linker conformations if they are in low abundance and/or classify into a limited number of 2D views thereby precluding the calculation of interpretable 3D EM maps.

We decided to analyse Rea1 linker remodelling by negative stain electron microscopy (EM). Compared to cryoEM, negative stain EM offers the advantage of increased signal-to-noise ratios for individual particles, which should ease the detection of low abundance Rea1 linker conformations. We reasoned that even though the resolution of negative stain EM is limited, large-scale remodelling events of the ≈200 kDa linker domain should still be detectable. We also limited our image processing workflow at the 2D classification stage to avoid failing to detect alternative linker conformations during 3D classification due to insufficient 2D projection distributions.

First, we characterized wild-type *S. cerevisiae* Rea1 (Rea1$_{wt}$) in the presence of ATP. We were able to obtain a subset of 2D classes with identical top views onto the AAA+ ring. In these 2D classes, the linker samples a range of different conformations with respect to the AAA+ ring (Fig. 1e). We tentatively sorted them into "Extended", "Intermediate" and "AAA+ ring engaged" linker conformations and assigned the numbers 1–7 from the most extended to the most compact Rea1 linker state (Fig. 1e). States 6 and 7, which represent the AAA+ ring engaged linker conformations, are clearly minority classes representing only ≈0.8% of the total particles in the data set. States 1–3 of the extended and state 4 of the intermediate linker conformations are similar to Rea1 linker conformations described in earlier studies[10] (Supplementary Fig. 1). In the nucleotide free APO data set, the AAA+ ring top view particles exclusively sorted into the extended and

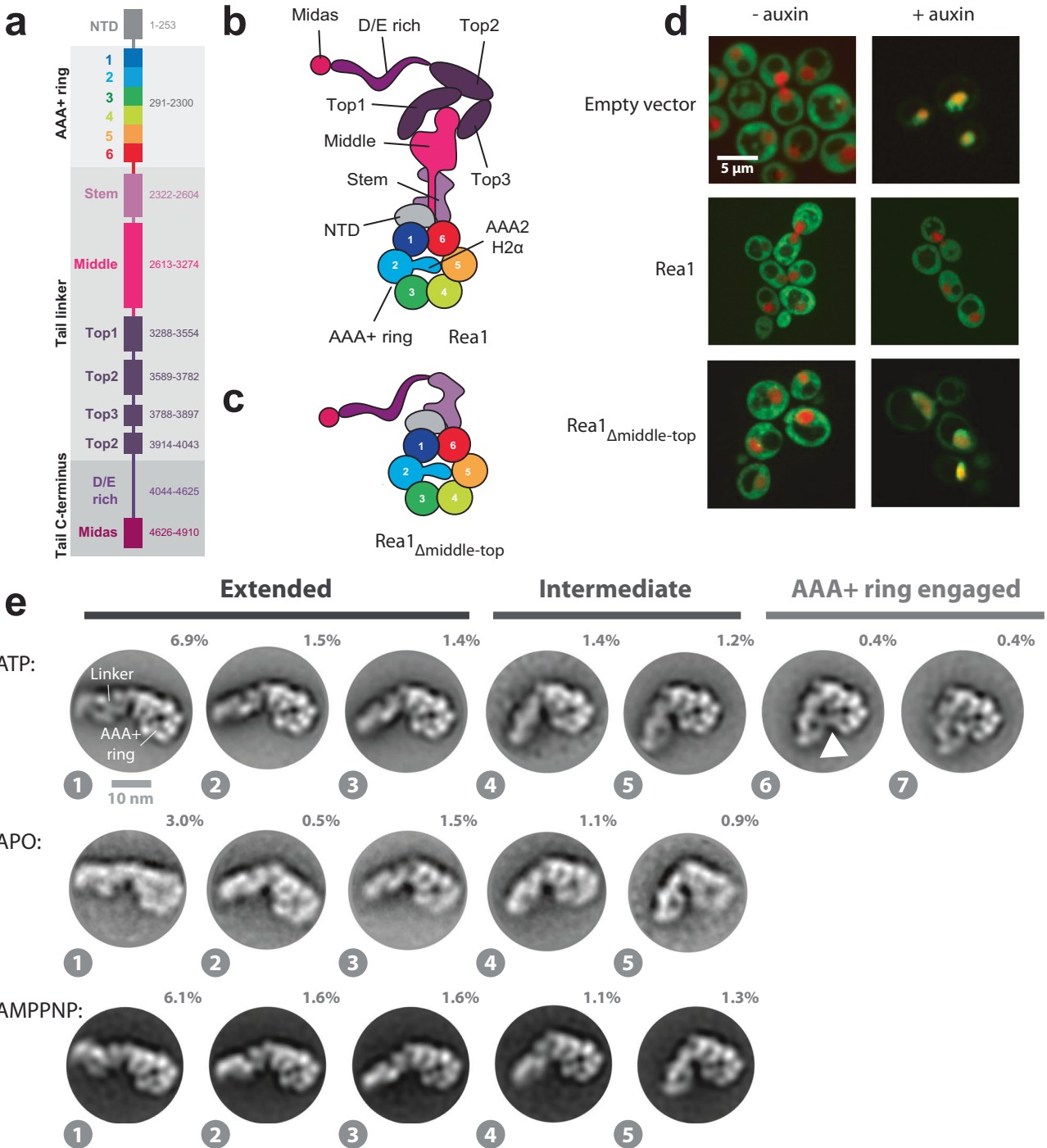

**Fig. 1 | The Rea1 linker is a functionally important structural element and shows nucleotide independent as well as nucleotide dependent steps of remodelling. a** Domain organization of Rea1. **b** Schematic cartoon representation of Rea1. The helix 2 α-helical insertion of the AAA2 domain (AAA2H2α) sits in the central pore of the AAA+ ring. Adapted from ref. 17. **c** Schematic cartoon representation of a construct lacking the linker middle and top domains. The flexible D/E rich region with the substrate binding MIDAS domain are directly fused to the linker stem domain. Adapted from Sosnowski et al.[17]. Elife 7, https://doi.org/10.7554/eLife.39163 (2018) under a CC BY license: https://creativecommons.org/licenses/by/4.0/. **d** Yeast nuclear export assay of pre-ribosomal particles. The pre-ribosomal particle marker Rpl25 is fused to GFP. Histone-mCherry marks the nucleus. The endogenous Rea1 is under the control of the auxin degron system. Upper panels: The addition of auxin leads to the accumulation of GFP fluorescence in the nucleus indicating a pre60S nuclear export defect due to degraded endogenous Rea1. Middle panels: The export defect can be rescued by a plasmid harbouring a Rea1_{wt} copy. Lower panels: Providing a plasmid harbouring the construct in C. does not rescue the export defect, suggesting the linker middle and top domains are functionally important. Experiments were repeated three times. **e** Negative stain 2D classes representing AAA+ ring top views of Rea1_{wt} in the presence of ATP, absence of nucleotide as well as in the presence of AMPPNP. States 1 – 5 represent the extended and intermediate linker conformations, which do not require nucleotide. In contrast to states 1 – 5, the AAA+ ring engaged states 6 and 7 require ATP hydrolysis. The white arrow head highlights a connection between the linker top and the AAA+ ring. Percentage numbers indicate how many particles of the corresponding data set sorted into the displayed 2D class averages.

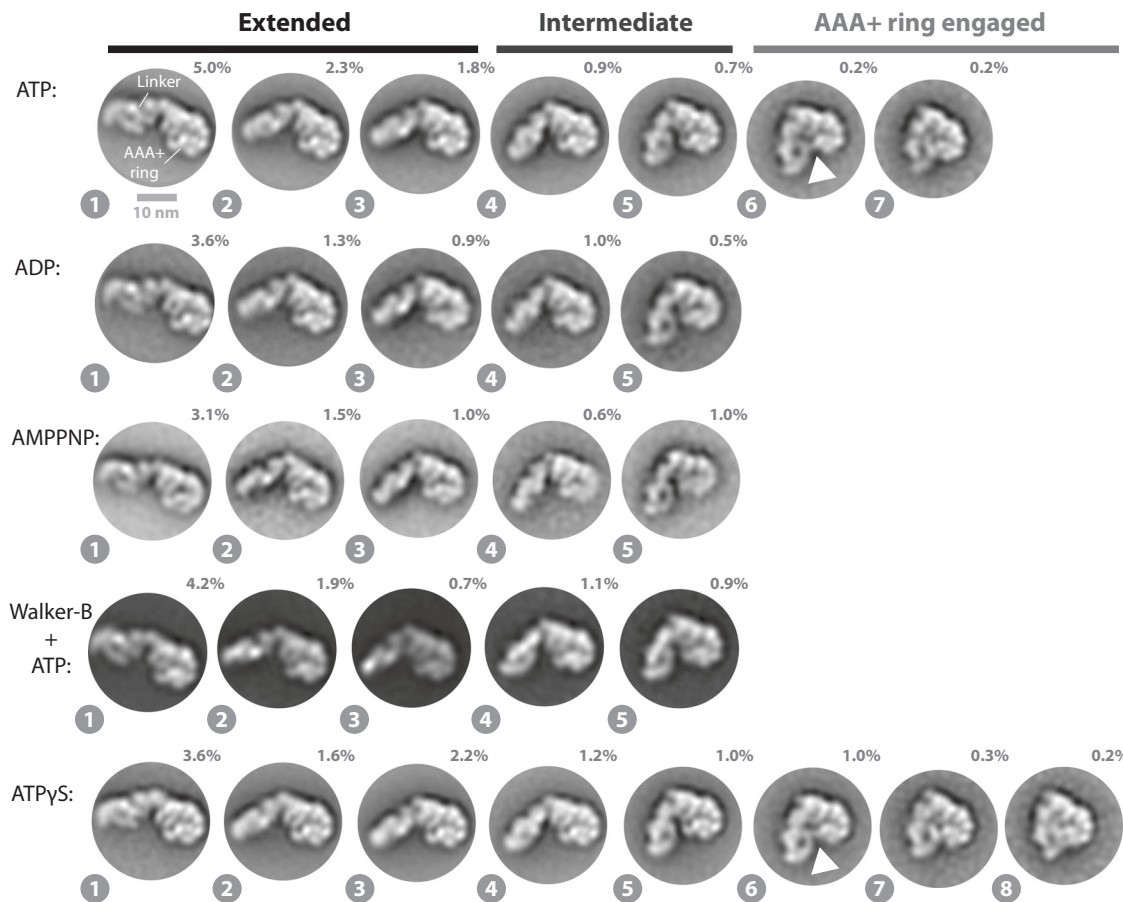

**Fig. 2 | Linker remodelling in Rea1$_{\Delta AAA2H2\alpha}$.** The analysis of the ATP, ADP, AMPPNP and ATPγS data sets indicates that linker remodelling in Rea1$_{\Delta AAA2H2\alpha}$ is highly similar to linker remodelling in Rea1$_{wt}$. Like in the case of Rea1$_{wt}$, ATP-hydrolysis is required to engage the linker with the AAA+ ring. The latter point is also supported by the Rea1$_{\Delta AAA2H2\alpha}$ Walker-B mutant, which is impaired in ATP-hydrolysis activity and does not show the AAA+ ring engaged 2D class averages. Unlike Rea1$_{wt}$, Rea1$_{\Delta AAA2H2\alpha}$ is able to sample state 8 in the presence of the slowly hydrolysable ATP analogue ATPγS. White arrow heads highlight a connection between the linker top and the AAA+ ring. Percentage numbers indicate how many particles of the corresponding data set sorted into the displayed 2D class averages.

intermediate linker conformations (states 1–5) (Fig. 1e). To rule out the possibility that some of the observed linker conformations under APO conditions are stabilized by co-purified nucleotides, we also carried out negative stain EM investigations on Rea1 samples purified in the presence of EDTA (Supplementary Fig. 2). We again only observed the extended and intermediate linker conformations, suggesting that the AAA+ ring engaged linker conformations require the presence of nucleotide. To further investigate whether ATP binding or hydrolysis is needed to engage the linker with the AAA+ ring, we collected a data set in the presence of the ATP analogue AMPPNP, which cannot be hydrolysed by Rea1 (Supplementary Fig. 3). Again, only the extended and intermediate linker conformations were sampled (states 1–5) (Fig. 1e) indicating that linker AAA+ ring engagement requires ATP-hydrolysis in the Rea1 AAA+ ring.

Next, we investigated a Rea1 mutant lacking the AAA2 helix 2 α-helical insert (AAA2H2α) (Rea1$_{\Delta AAA2H2\alpha}$) (Fig. 1b). In a previous study, we demonstrated that AAA2H2α is an auto-inhibitory regulator of the Rea1 ATPase activity that also prevents the docking of the MIDAS domain onto the AAA+ ring[17]. The relocation of AAA2H2α from the central pore of the AAA+ ring and the docking of the MIDAS domain onto the AAA+ ring is also observed when Rea1$_{wt}$ is bound to Rsa4-pre60S particles[19,24]. This suggests that the Rea1$_{\Delta AAA2H2\alpha}$ mutant resembles Rea1$_{wt}$ when bound to pre60S particles.

We determined negative stain EM 2D class averages of the Rea1$_{\Delta AAA2H2\alpha}$ mutant in the ATP, APO, ADP and AMPPNP states. In the presence of ATP, the AAA+ ring top view particles sorted again in extended, intermediate as well as AAA+ ring engaged classes (Fig. 2). The linker conformations are highly similar to the ones observed for Rea1$_{wt}$ (Fig. 1e), suggesting that linker remodelling is not altered in Rea1$_{\Delta AAA2H2\alpha}$. In the absence of nucleotide, we detected linker conformations that are consistent with states 1–5 of the extended and intermediate classes (Supplementary Fig. 4). However, the AAA+ ring appeared to be partially flexible suggesting that in Rea1$_{\Delta AAA2H2\alpha}$ nucleotide is required to stabilize the AAA+ ring. Incubating Rea1$_{\Delta AAA2H2\alpha}$ with ADP or AMPPNP, which cannot be hydrolysed by Rea1$_{\Delta AAA2H2\alpha}$ (Supplementary Fig. 3), restricted linker remodelling to the extended and intermediate classes (Fig. 2). The results for AMPPNP again suggest that ATP binding is not sufficient to engage the linker with the AAA+ ring. Following an established strategy in the AAA+ field, we also investigated the Rea1$_{\Delta AAA2H2\alpha}$ Walker-B mutant in the presence of ATP to further strengthen this point. Mutating the catalytic Walker-B glutamates in the AAA+ ring to glutamines impairs the ATPase activity of AAA+ proteins resulting in the stabilization of the ATP state. Also under these conditions, we only detected the extended and intermediate classes (Fig. 2). Like in the case of Rea1$_{wt}$, ATP-hydrolysis in the Rea1$_{\Delta AAA2H2\alpha}$ AAA+ domains drives the engagement of the linker with the AAA+ ring. AAA+ ring engaged classes were again minority views representing only ≈0.4% of the total particles.

Recent cryoEM studies on a AAA+ unfoldase made use of the slowly hydrolysable ATP analogue ATPγS to enrich transient protein conformations[25]. We tested if ATPγS, which is also slowly hydrolysed by Rea1$_{\Delta AAA2H2\alpha}$ (Supplementary Fig. 3), might stabilize the AAA+ ring

engaged linker states in Rea1$_{\Delta AAA2H2\alpha}$. Although we did not observe a substantial enrichment (≈ 0.4% ATP vs. ≈1.5% ATPγS), we detected state 8, an additional AAA+ ring engaged linker conformation (Fig. 2). We also investigated, if state 8 is sampled in Rea1$_{wt}$ in the presence of ATPγS. Like in the case of Rea1$_{\Delta AAA2H2\alpha}$, Rea1$_{wt}$ slowly hydrolyses ATPγS (Supplementary Fig. 3). We detected 2D class averages similar to state 8 of Rea1$_{\Delta AAA2H2\alpha}$, but with less well-defined structural features of the linker indicating increased structural flexibility (Supplementary Fig. 5).

Collectively, these results demonstrate that distinct Rea1 linker remodelling steps exist. The extended and intermediate linker conformations (states 1–5) are being sampled even in the absence of any nucleotide highlighting the intrinsic conformational flexibility of the linker. The AAA+ ring engagement of the linker requires ATP-hydrolysis in the Rea1 AAA+ ring, which suggests an important functional role for these low abundance conformations. This view is also supported by Rsa4 and Ymt1 in-vitro release assays, which demonstrated that only ATP but not AMPPNP enables Rea1 to catalyse the removal of assembly factor substrates from pre60S particles[8,10]. These findings indicate that the extended and intermediate linker conformations (states 1–5) (Figs. 1e and 2) are insufficient to support functionality and that additional conformations linked to ATP-hydrolysis (states 6-8) have to be sampled to catalyse assembly factor removal. Linker remodelling of Rea1$_{\Delta AAA2H2\alpha}$ and Rea1$_{wt}$ are largely similar, but Rea1$_{\Delta AAA2H2\alpha}$ is able to stably sample state 8, unlike Rea1$_{wt}$.

## The Linker top and middle domains rotate and pivot towards the AAA+ ring docked MIDAS domain during remodelling

Since Rea1$_{\Delta AAA2H2\alpha}$ in the ATPγS state revealed the highest number of linker remodelling conformations, we thought to annotate the Rea1 subdomains in the corresponding 2D class averages to analyse Rea1 linker remodelling in more detail. To this end, we determined a Rea1$_{\Delta AAA2H2\alpha}$ ATPγS 3D cryoEM structure (Fig. 3a, b, Supplementary Figs. 6 and 7, and Supplementary Table 1) to generate 2D projections for the subdomain assignment in our negative stain EM 2D classes. Consistent with previous cryoEM investigations[17], we could only obtain an interpretable 3D reconstruction showing the linker in the straight conformation, which was the dominant structural state in the sample and already described in earlier work (Supplementary Fig. 8)[17,19,20]. Consistent with the cryoEM data, the straight linker conformation is also the dominant structural state in the Rea1$_{\Delta AAA2H2\alpha}$ ATPγS negative stain EM data set (Supplementary Fig. 9).

We generated a 2D projection from the cryoEM map with high similarity to the AAA+ ring in our negative stain EM 2D class averages (Fig. 3c), but the linker part did not resemble any of the observed linker conformations of the eight states (Fig. 2). Using a different orientation, we were able to obtain a 2D projection matching the linker of state 1 but not the AAA+ ring (Fig. 3c). The combination of both projections allowed us to assign the NTD, the six AAA+ domains of the AAA+ ring as well as the linker stem, middle and top domains and the MIDAS domain in state 1 (Fig. 3c). Our analysis also demonstrates that the linker in state 1 has already undergone a rigid-body movement with respect to the AAA+ ring compared to the straight linker conformation in our cryoEM structure. We approximate that the linker top and middle domains have rotated ≈30 ° counter clock wise around the long linker axis and bent ≈ 45 ° towards the plane of the AAA+ ring (Supplementary Fig. 10).

A prominent feature of the Rea1$_{\Delta AAA2H2\alpha}$ AAA+ ring in states 1–8 is a bright spot on the AAA+ ring, which we interpret as AAA+ ring docked MIDAS domain (Fig. 3c). This feature is absent in the Rea1$_{wt}$ classes (Fig. 3d and Supplementary Fig. 11), which is consistent with earlier findings demonstrating that the presence of the AAA2 H2 α-helical insert in Rea1$_{wt}$ interferes with the AAA+ ring docking of the MIDAS domain[17]. We also directly confirmed our MIDAS domain assignment by analysing the Rea1$_{\Delta AAA2H2\alpha-\Delta MIDAS}$ double mutant (Fig. 3d).

Next, we aligned the eight states onto their AAA+ rings (Fig. 4a and Supplementary Movie 1). These AAA+ ring alignments suggest that the linker top and middle domains pivot towards the AAA+ ring docked MIDAS domain during linker remodelling. The pivot point during this movement is located between the linker middle and stem domains. The alignments further suggest that the linker top and middle domains rotate during the pivot movement. To better visualize this additional transformation, we aligned states 1–8 on the long linker axis (Fig. 4b and Supplementary Movie 2) to demonstrate the rotation, which occurs during states 1–5. The linker top and middle domains behave approximately as a rigid body during the rotation as suggested by a series of 2D projections of the linker top and middle domains cryoEM map rotated around the long linker axis (Fig. 4c, d). We estimate the rotation angle between states 1 and 5 to be ≈100 ° (Fig. 4e). In the final linker remodelling conformation, state 8, the linker top2 and top3 domains are in close proximity to the AAA+ ring docked MIDAS domain (Fig. 4a, b, and Supplementary Movies 1 and 2).

These results reveal that the linker top and middle domains undergo a complex series of movements with respect to the linker stem domain and the AAA+ ring during linker remodelling. Compared to the straight linker conformation in Fig. 3a, they rotate and pivot towards the plane of the AAA+ ring in an initial movement to reach state 1. From the position in state 1 they pivot towards the AAA+ ring docked MIDAS domain and further rotate around the long linker axis. The rotation largely happens during states 1–5, which cover the extended and intermediate linker conformations. As these states are also observed under APO conditions (Fig. 1e and Supplementary Figs. 2 and 4), we conclude that the rotational movement and the additional swing towards the AAA+ ring do not require energy, suggesting they are part of the intrinsic conformational flexibility of the Rea1 linker. The energy provided by ATP-hydrolysis in the AAA+ ring is needed to engage the fully rotated linker top and middle domains with the AAA+ ring during states 6–8. The connection between the linker and the AAA+ ring in state 6 that separates the AAA+ ring engaged conformations from the intermediate conformations occurs between the linker top3 domain and AAA1S (Supplementary Fig. 12). In state 8, the linker top2 and top3 domains are in close proximity to the AAA+ ring docked MIDAS domain (Fig. 4a, b). The fact that state 8 was not observed in Rea1$_{wt}$, which does not show the AAA+ ring docked MIDAS domain (Fig. 3d and Supplementary Fig. 11), suggests that the presence of the MIDAS domain at the AAA+ ring is essential for sampling state 8.

## The Linker top interacts with the MIDAS domain and linker remodelling is a force producing event

Since the linker top2 and top3 domains in state 8 are located next to the MIDAS domain, we tested if there is a direct interaction between these two regions of the protein. To this end, we carried out cross-linking mass spectrometry (XL-MS) using PhoX on the Rea1$_{\Delta AAA2H2\alpha}$ mutant in the presence of ATPγS and identified 182 K-K crosslinks (Supplementary Fig. 13a, b, c and Supplementary Table 2). Highlighting the good quality and specificity of the XL-MS data, 77 out of the 182 K-K crosslinks (42%) could be assigned and distance-validated to the straight linker conformation shown in Fig. 3a, which is the expected dominant structural state in the sample.

Although our negative stain EM analysis suggested that state 8 only represents ≈ 0.2% of the total particles in our sample, PhoX-enabled cross-links enrichment allowed the detection of a crosslink between K3955 in the top2 domain and K4662/K4668 in the MIDAS domain (Fig. 5a). An additional crosslink was identified between K3569, located in the linker top1-top2 loop region that associates with the top2 domain[17], and K4662. The K3569-K4662 and K3955-K4662/K4668 crosslinks further support our domain assignments in state 8 (Fig. 4a, b) and hint at a direct interaction between the linker top and the MIDAS domain. Consistent with our negative stain EM analysis, we did not detect these crosslinks in the absence of nucleotide or

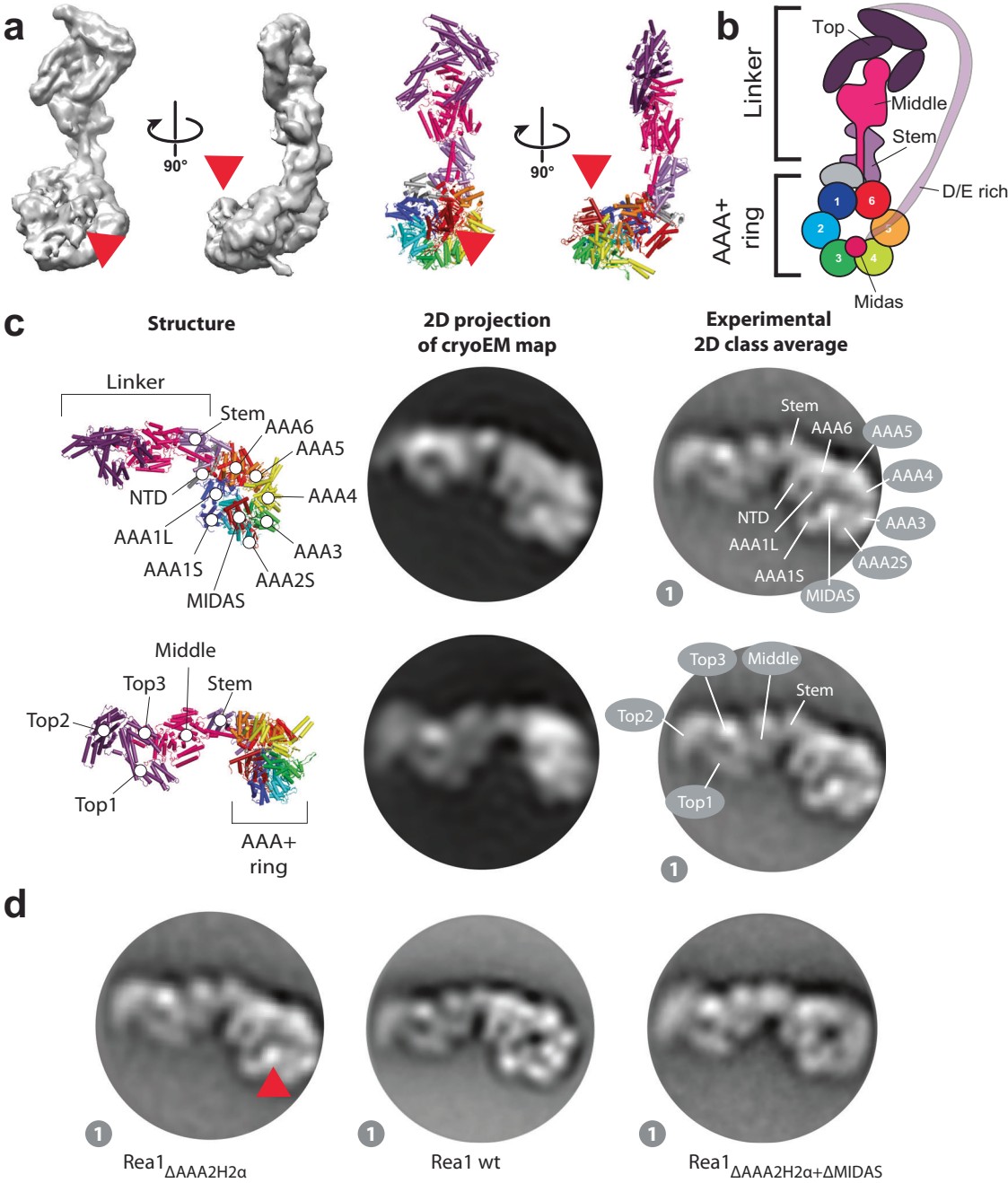

**Fig. 3 | Domain assignments in negative stain 2D class averages of Rea1$_{\Delta AAA2H2\alpha}$. a** CryoEM map (left panels) and cartoon representation (right panels) of Rea1$_{\Delta AAA2H2\alpha}$ in the presence of ATPγS. The red arrow heads highlight the AAA+ ring docked MIDAS domain. **b** Schematic cartoon representation of the structure in a. The α-helical extension of AAA2 normally occupying the centre of the pore (compare Fig. 1b) has been deleted, which allows the MIDAS domain to dock onto the AAA+ ring. The D/E rich region connecting the MIDAS domain to the linker top is flexible and not visible in the structure. Adapted from Sosnowski et al.[17]. Elife 7, https://doi.org/10.7554/eLife.39163 (2018) under a CC BY license: https://creativecommons.org/licenses/by/4.0/. **c** Upper panels: A 2D projection of the cryoEM map in A. low pass filtered to 25 Å shows a good match for the AAA+ ring region in the state 1 negative stain 2D class average. The projection allows the assignment of the NTD, AAA1–AAA6, the linker stem and the MIDAS domain. In

contrast to the AAA+ ring region, the linker top and middle domains adopt a different conformation from the one seen in state 1. Lower panels: With a different 2D projection of the low pass filtered cryoEM map, a good match for the linker region in the state 1 negative stain 2D class average can be produced, allowing the assignment of the linker top1, top2 and top3 domains as well as the linker middle domain. The AAA+ ring does not match up with the AAA+ ring in the state 1 negative stain 2D class average. This mismatch indicates that -compared to the cryoEM structure in A. - the linker top and middle domains in state 1 have moved with respect to the AAA+ ring. **d** The assignment of the MIDAS domain in state 1 of Rea1$_{\Delta AAA2H2\alpha}$ (red arrow head, left panel) is further supported by comparisons with state 1 of Rea1$_{wt}$, where the MIDAS domain is absent from the AAA+ ring (middle panel, compare also Supplementary Fig. 11) as well as analysis of the Rea1$_{\Delta AAA2H2\alpha+\Delta MIDAS}$ double mutant (right panel).

presence of AMPPNP (Supplementary Fig. 13a, c and Supplementary Table 2).

The top2 domain crosslink partners K4662 and K4668 are located in the highly conserved E4656-K4700 loop of the MIDAS domain,

which is not involved in Ytm1 or Rsa4 assembly factor binding[21]. This loop region harbours an NLS sequence that is required for the nuclear import of Rea1[21]. It was also demonstrated that the E4656-K4700 loop is essential for assembly factor removal. Deleting and replacing it with

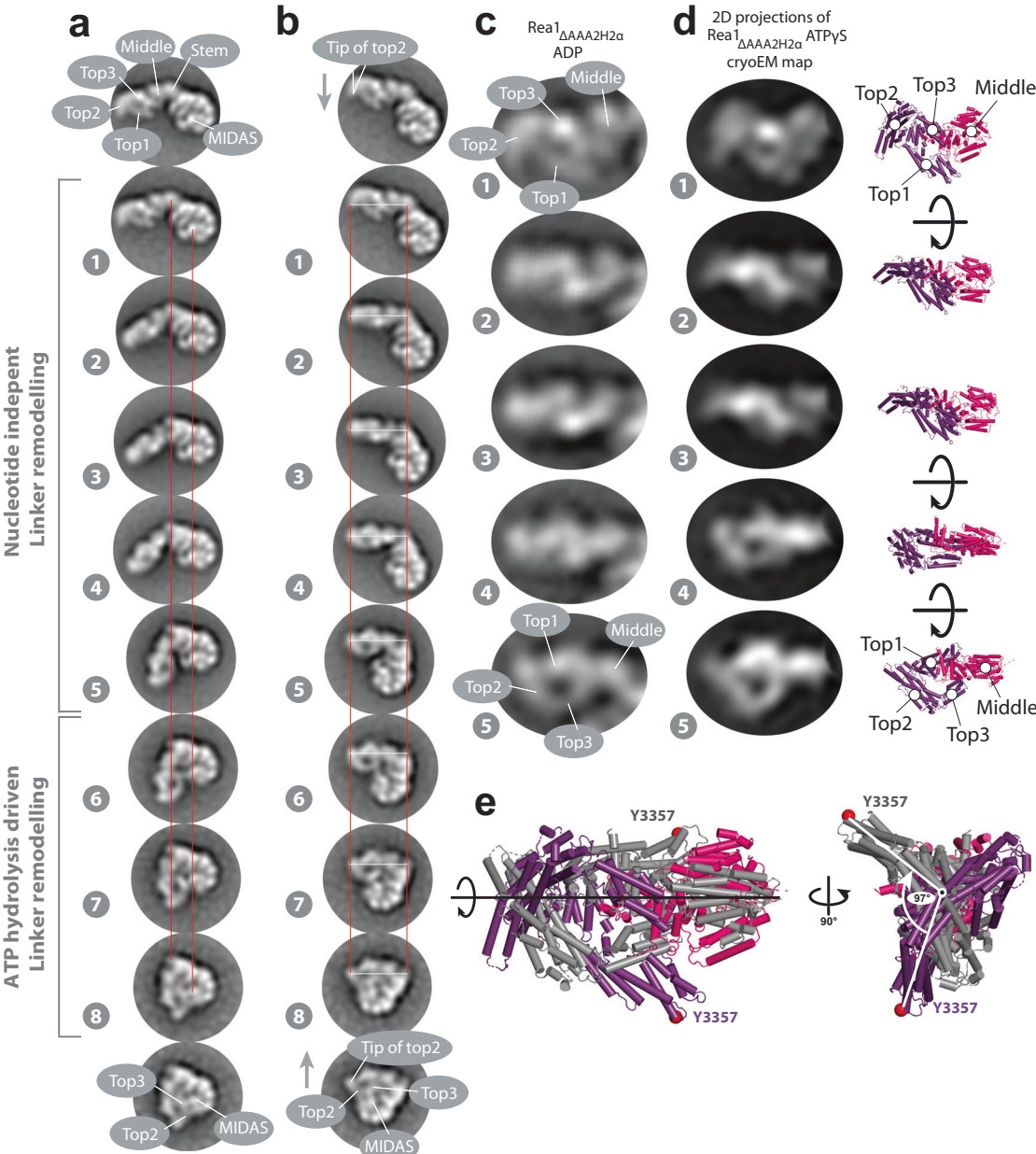

**Fig. 4 | The Rea1 linker pivots and rotates during remodelling. a** States 1 – 8 of Rea1$_{\Delta AAA2H2\alpha}$ ATPγS aligned on the linker stem and MIDAS domains (red lines). The linker middle and top domains swing towards the AAA+ docked MIDAS domain (compare also Supplementary Movie 1). The region between the linker stem and middle domains acts as pivot point. **b** States 1–8 of Rea1$_{\Delta AAA2H2\alpha}$ ATPγS aligned on long linker axis (tip of linker top2 domain and linker stem domain, white and red lines). The linker top and middle domains rotate during the pivot swing (compare also movie S2). The tip of the linker top2 domain points downwards in state 1 (grey arrow) but upwards in state 8 (grey arrow) highlighting the rotation. In the final linker remodelling conformation, state 8, the linker top2 and top3 domains as well as the MIDAS domain are in close proximity. States 1 -5 were also observed under APO conditions (compare Fig. 1e) indicating that large parts of the linker swing and

the linker rotation are nucleotide independent and are part of the intrinsic conformational flexibility of the linker. The engagement of the rotated linker with the AAA+ ring during states 6–8 requires ATP hydrolysis. **c** Enlarged views of linker region in states 1–5 of Rea1$_{\Delta AAA2H2\alpha}$ ADP. **d** Left panels: Series of 2D projections of the linker top-middle part of the Rea1$_{\Delta AAA2H2\alpha}$ ATPγS cryoEM map low pass filtered to 25 Å and rotated around the long linker axis. Right panels: corresponding structures. The rotation of the linker in a and b can be approximated by a rigid-body rotation of the linker top and middle domains around the long linker axis. Additional internal rearrangements of the linker middle and top domains with respect to each other cannot be excluded. **e** Aligning states 1 (color coded) and 5 (grey) of d on the long linker axis indicates a total rotation angle of ≈100°. Equivalent Cα atoms are shown as red spheres.

an alternative NLS sequence rescued nuclear import but prevented Rea1 from removing its Rsa4 assembly factor substrate from pre60S particles[21]. These results suggest that the E4656-K4700 loop plays an indirect role during assembly factor removal.

To provide additional support for an interaction between the linker top and the MIDAS domain, we carried out yeast-two-hybrid assays, but the expression of the fusion constructs turned out to be

toxic for the cells (Supplementary Fig. 14a). However, a subsequent immunoprecipitation of the MIDAS domain construct provided evidence for an interaction with the linker top2/top3 domains (Supplementary Fig. 14b). We also carried out in-vitro pulldown experiments as additional validation for the linker top2/top3–MIDAS domain interaction. While we were not able to express the corresponding *S. cerevisiae* Rea1 domains in bacteria, we succeeded in producing their

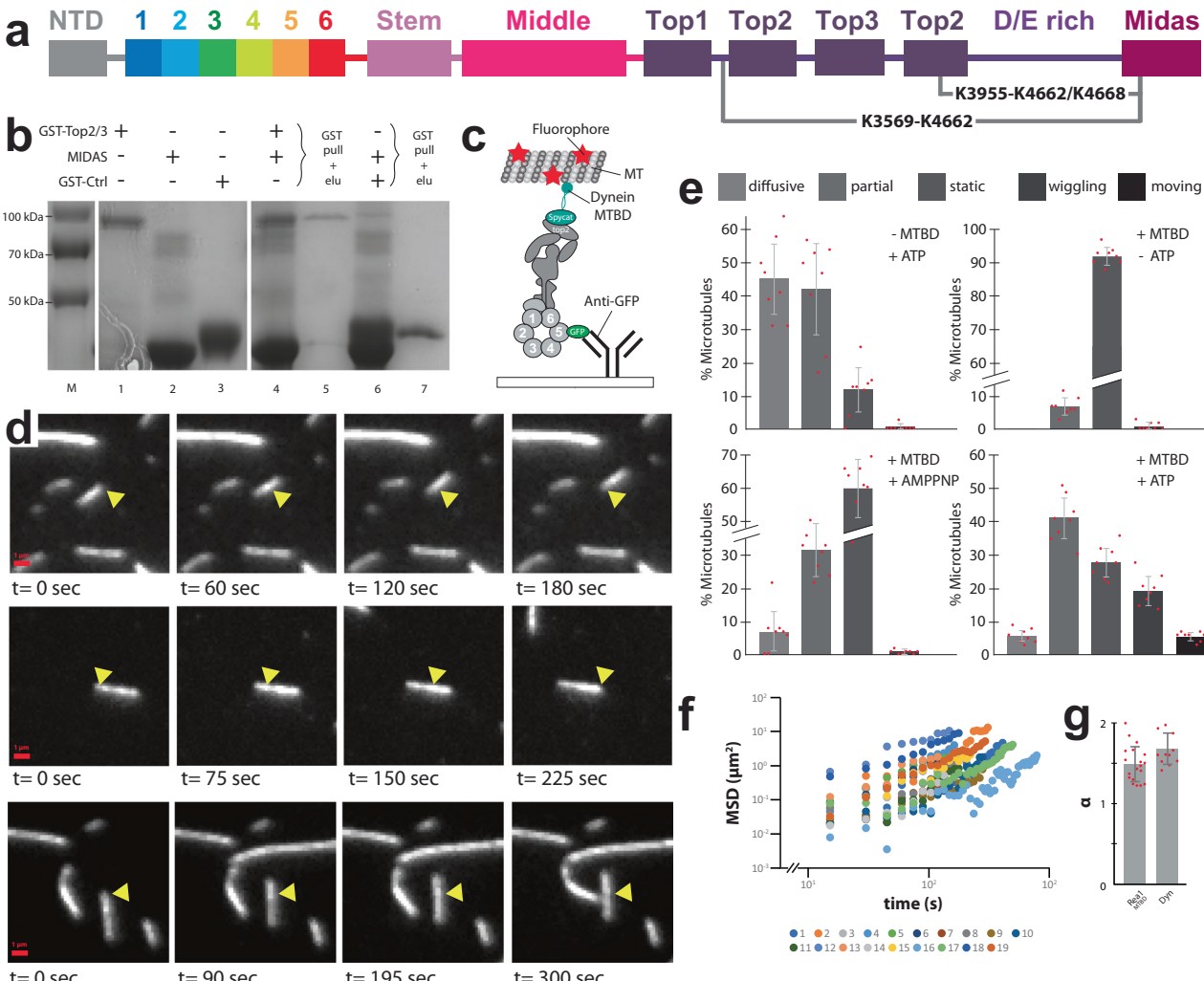

**Fig. 5 | The linker top interacts with the MIDAS domain and linker remodelling is a force producing event. a** Crosslinks supporting state 8 detected in the Rea1$_{\Delta AAA2H2\alpha}$ ATPγS data set. The K3569-K4662 and K3955-K4662/K4668 crosslinks suggest an interaction between the linker top2 domain and a conserved MIDAS domain loop. **b** GST pulldown experiments provide further support for the interaction between the MIDAS and linker top2/3 domains. The samples were run on a SDS gel and stained with Coomassie blue. The GST-linker top2/3 fragment pulls down the MIDAS domain (lane 5). M: marker, lanes 1, 2 and 3: purified linker top2/3 construct, MIDAS domain and GST control. Lane 4, 5: GST-linker top2/3 + MIDAS domain input and pulldown. Lane 6, 7: GST-control + MIDAS domain input and pulldown. The pulldown experiment was carried out once. **c** Microtubule gliding assays provide evidence that Rea1 linker remodelling produces force. The dynein microtubule binding domain (cyan) was fused to the linker top2 domain using the spycatcher/spytag approach. GFP was fused to AAA5. The construct was anchored to a cover slide via GFP-antibodies and fluorescently labelled microtubules we

applied. Adapted from Sosnowski et al.[17] Elife 7, https://doi.org/10.7554/eLife.39163 (2018) under a CC BY license: https://creativecommons.org/licenses/by/4.0/. **d** Movie frames of microtubule gliding events. Yellow arrow heads mark the microtubule position at the beginning of the movie (Supplementary Movies 3–5). The gliding events suggest that linker remodelling with respect to the AAA+ ring is able to produce mechanical force. **e** Statistical analysis of diffusive, partially bound, static, wiggling and moving microtubules under different conditions ($n = 8$). Moving microtubules are only observed in the presence of the dynein microtubule domain (MTBD) and ATP. **f, g** The microtubule gliding events result from directed movement. **f** Log mean squared displacement vs log time. **g** Slopes of the events in f, which represent the anomalous coefficient α, were averaged ($n = 19$). The average α of 1.49 ± 0.22 indicates that the events are not caused by a random diffusion process (α = 1). The motor protein dynein, known to power directed microtubule gliding, has a comparable α (1.65 ± 0.18, $n = 11$). Error bars show the standard deviation. Source data are provided as a Source Data file.

*C. thermophilum* homologues. The results of the GST-pulldown experiments confirmed the interaction between the linker top2/top3 and MIDAS domains (Fig. 5b). The interaction with the linker top2/top3 domain was not detected for a MIDAS domain construct lacking the conserved T4733-K4778 loop, which is equivalent to the conserved *S. cerevisiae* MIDAS E4656-K4700 loop[21] (Supplementary Fig. 15).

In order to probe if linker remodelling is able to produce mechanical force, we carried out total-internal-reflection-fluorescence (TIRF) microscopy based microtubule gliding assays. We worked with the Rea1$_{\Delta AAA2H2\alpha}$ background since this construct shows elevated ATPase activity compared to the wt[17]. We modified the construct further to adapt it for the microtubule gliding assays (for details see

"materials and methods"). Importantly, we introduced a spytag inside the linker top2 domain to which we covalently attached the spycatcher labelled dynein microtubule binding domain (Fig. 5c). After the application of fluorescently labelled microtubules, we screened for microtubule gliding events. Since force production for microtubule gliding relies on energy provided by ATP-hydrolysis and our negative stain EM analysis suggests that only a minority of Rea1 molecules adopt the ATP-hydrolysis associated conformations 6, 7 and 8 in the presence of ATP, microtubule gliding is expected to be a rare event. Consistent with this assumption, we observed occasional events of slow, directed microtubule gliding over several minutes (Fig. 5d, e and Supplementary Movies 3–5). We did not observe such events in the absence of

nucleotide, for constructs without dynein microtubule binding domain or in presence of AMPPNP (Fig. 5e). To further rule out that the microtubule gliding events we detected result from random diffusion, we additionally carried out mean-squared displacement analyses (Fig. 5f, g). These results indicate that the linker remodelling states driven by ATP-hydrolysis in the AAA+ ring are able to produce mechanical force.

Taken together, our results suggest that the mechanical force produced by linker remodelling might be directly applied to the AAA+ ring-docked MIDAS domain via the linker top2/top3 domains to remove assembly factors from pre60S particles. The conserved E4656-K4700 loop of the MIDAS domain participates in the linker top2/3 - MIDAS interaction.

### A highly conserved salt-bridge network in the linker middle domain is crucial for linker remodelling

Next, we were searching for structural elements in the linker that might be important for its remodelling. To this end, we screened for highly conserved amino-acid residues in the Rea1 linker. Although the general conservation within the Rea1 linker domain is low, we nevertheless identified a highly conserved salt-bridge network, D2915-R2976-D3042, in close proximity to the α-helical extension of the linker middle domain (Fig. 6a, b, c). The disruption of this salt-bridge network by alanine mutations changed the remodelling pathway of the linker. While the region between the linker middle and stem domains still acts as pivot point for the swing of linker top and middle domains towards the AAA+ ring, Rea1$_{D2915A-R2976A-D3042A}$ did not show the rotation around the long linker axis that we consistently detected in all previous data sets (Fig. 6d and Supplementary Movie 6). Consequently, the linker is not able to engage with the AAA+ ring. The inability of the Rea1$_{D2915A-R2976A-D3042A}$ linker to correctly engage with the AAA+ ring in the presence of ATP also affects the ATPase activity of this construct (Fig. 6e). These defects suggest that linker engagement with the AAA+ ring in Rea1$_{wt}$ stimulates the ATPase activity.

We also characterized Rea1$_{D2915A-R2976A-D3042A}$ by cryoEM in the presence of ATP. We were able to obtain three reconstructions at medium to low resolution (Fig. 6f, Supplementary Figs. 16 and 17, and Supplementary Table 1). As expected, the straight linker was the dominant class (conformation I) (Supplementary Fig. 8e). It refined to a resolution sufficient to resolve secondary structure elements. In addition, we were able to resolve two alternative linker conformations of Rea1$_{D2915A-R2976A-D3042A}$ (conformations II and III) at a resolution allowing the docking of the linker stem-AAA+ ring and the linker top-middle domains. As expected, conformations II and III were related by a swing of the linker top and middle domains towards the AAA+ ring without rotation around the long linker axis (Supplementary Fig. 18). These 3D cryoEM reconstructions confirmed our initial assignment of the linker middle-stem region as pivot point for linker remodelling.

To assess the functionality of the Rea1$_{D2915A-R2976A-D3042A}$ mutant, we first tested its ability to support growth in the absence of endogenous wild-type Rea1. A centromeric plasmid directing expression of Rea1$_{D2915A-R2976A-D3042A}$ from the *REA1* promoter was transformed in a heterozygous *REA1/rea1::kanR* diploid strain bearing a deletion of one *REA1* allele. Following sporulation and tetrad dissection, *rea1::kanR* haploids expressing Rea1$_{D2915A-R2976A-D3042A}$ could be obtained (Fig. 7a). However, they were very slow growing, underscoring the functional importance of the conserved salt-bridge network. We verified that the slow growth phenotype is not due to lower expression levels of the Rea1$_{D2915A-R2976A-D3042A}$ mutant (Supplementary Fig. 19). We next assessed the effect of the Rea1$_{D2915A-R2976A-D3042A}$ mutations on pre-rRNA processing by northern analyses (Fig. 7b). We performed a first analysis using a *GAL::rea1* strain conditionally expressing endogenous Rea1, transformed with the plasmid mentioned above. The strain was first propagated in galactose-containing medium and shifted to glucose-containing medium to repress wild-type Rea1

expression. The northern analyses indicated that the Rea1$_{D2915A-R2976A-D3042A}$ mutant leads to similar pre-rRNA processing defects as the lack of Rea1, characterized by an increase in the levels of early 35S pre-rRNA and intermediate 27SB pre-rRNA, reflecting defective assembly of both early and intermediate pre-ribosomal particles (Fig. 7b and Supplementary Fig. 20). We next carried out a more thorough analysis with the haploid *rea1::kanR* strain expressing the Rea1$_{D2915A-R2976A-D3042A}$ mutant. The salt bridge mutations induced an increase in the levels of the 7S pre-rRNA, the precursor to 5.8S rRNA, suggesting that later pre-60S particle maturation stages are also affected (Fig. 7c). As a result of the pre-rRNA processing defects, 25S rRNA levels were reduced in strains expressing Rea1$_{D2915A-R2976A-D3042A}$ (Fig. 7c).

We next investigated whether the pre-rRNA processing defects correlate with a perturbation in the association of Rea1 substrates, Ytm1 and Rsa4, with pre-60S particles. We immunoprecipitated HA-tagged versions of Ytm1 and Rsa4 and analysed the co-immunoprecipitated pre-rRNAs using northern analyses. To our surprise, we observed that Rea1 depletion does not have a major impact on the association of HA-tagged Ytm1 with 27S pre-rRNAs (Fig. 7d). However, Rea1 depletion leads to a significant drop in the co-precipitation efficiency of 27SB pre-rRNA with HA-tagged Rsa4 (Fig. 7d). We envisage that in the absence of Rea1, a failure to correctly remodel early Ytm1-containing pre-60S particles perturbs the later association of Rsa4 with incorrectly assembled intermediate pre-60S particles containing the 27SB pre-rRNA. We therefore then used HA-Rsa4 association with 27SB pre-rRNA as a readout of mutant Rea1 activity. In strains expressing Rea1$_{D2915A-R2976A-D3042A}$ the association of HA-Rsa4 with 27SB pre-rRNA was reduced relative to the wild-type control (Fig. 7e), but did not reach the level of the empty vector control, consistent with residual activity. Consistent with the pre-rRNA processing and Ytm1/Rsa4 release defects, Rea1$_{D2915A-R2976A-D3042A}$ also leads to a major inhibition of pre-60S particle export (Fig. 7f).

These results highlight the important role of the linker middle domain for Rea1 linker remodelling. Its conserved D2915-R2976-D3042 salt-bridge network is required for the correct rotation of the linker top and middle domains around the long linker axis during remodelling. Even though the long linker axis rotation is part of the intrinsic structural flexibility of the Rea1 linker and nucleotide independent, it nevertheless is of functional importance. The rotation during states 1 −5 is required to ensure the correct orientation of the linker top prior to its ATP-hydrolysis driven engagement with the AAA+ ring. The inability to correctly sample linker states 6, 7 and 8, which are associated with Rea1 AAA+ ring ATP-hydrolysis, correlates with functional defects in pre60S particle maturation and nuclear export.

## Discussion

The results presented here reveal the general principles of the Rea1 mechanism (Fig. 8), one of the largest and most complex ribosome maturation factors. We have demonstrated that the Rea1 linker is a functionally important structural element that undergoes an elaborated series of remodelling events. Two general aspects characterize and connect these remodelling events. The first aspect is the swing of the linker top and middle domains towards the AAA+ ring docked MIDAS domain with the region in-between the linker middle and stem domains acting as pivot point. The second aspect is the rotation of the linker top and middle domains around the long linker axis. A particularly interesting feature of this remodelling pathway is the occurrence of events that are part of the intrinsic structural flexibility of the linker like states 1 − 5. Even though these remodelling events do not depend on nucleotide binding or energy provided by ATP-hydrolysis, they still play an important functional role. They ensure that the linker top domains are in close proximity to the AAA+ ring and have the correct orientation prior to the ATP-hydrolysis driven engagement with the AAA+ ring. The latter aspect is particularly highlighted by the inability of the Rea1$_{D2915A-R2976A-D3042A}$ mutant to correctly rotate its linker

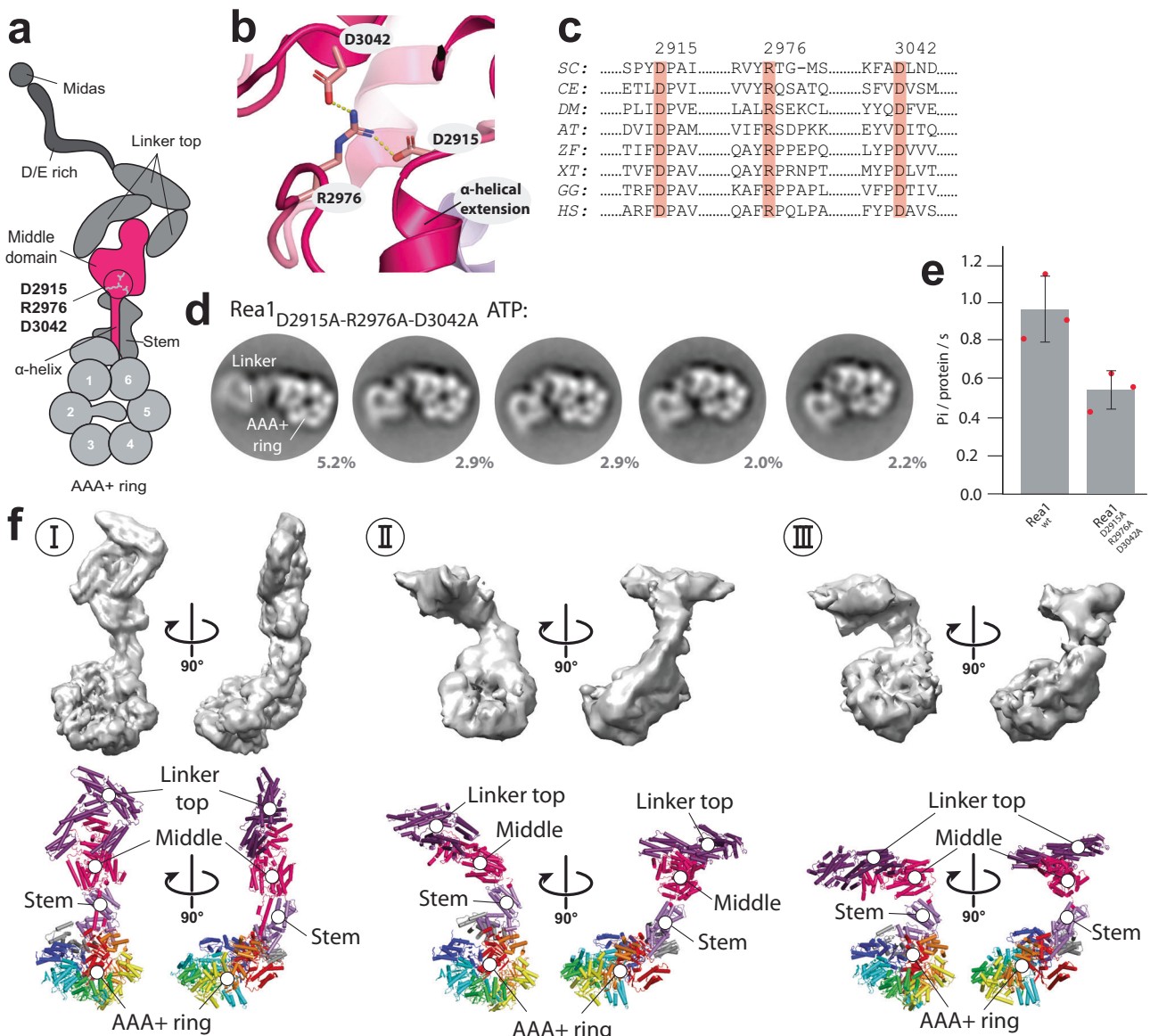

**Fig. 6 | A highly conserved salt-bridge network in the linker middle domain has essential functions during linker remodelling. a** Schematic representation of the highly conserved linker middle domain salt-bridge network D2915-R2976-D3042. It is located above the α-helical extension of the middle domain and at the interface between the linker middle and stem domains. Adapted from Sosnowski et al.[17]. Elife 7, https://doi.org/10.7554/eLife.39163 (2018) under a CC BY license: https://creativecommons.org/licenses/by/4.0/. **b** The D2915-R2976-D3042 salt-bridge network in the near-atomic resolution model of the *S. cerevisiae* Rea1 linker (PDB-ID: 6hyd). The dashed yellow lines indicate hydrogen bonds with a distance of 2.7 Å. **c** The D2915-R2976-D3042 salt-bridge network is conserved in Rea1/Midasin of *SC*: Saccharomyces cerevisiae, *CE*: Caenorhabditis elegans, *DM*: Drosophila melanogaster, *AT*: Arabidopsis thaliana, *ZF*: Zebrafish, *XT*: Xenopus tropicalis *GG*: Gallus gallus, *HS*: Homo sapiens. The *S. cerevisiae* Rea1 amino-acid residue numbering is shown at the top of the multiple sequence alignment. **d** Disrupting the D2915-R2976-D3042 salt-bridge network results in altered linker remodelling pathways. The linker is still able to swing towards the AAA+ ring but does not rotate around

the long linker axis, which prevents the engagement of the linker with the AAA+ ring (compare also Supplementary Movie 6). **e** The inability of the Rea1 linker top and middle domains to correctly engage with the Rea1 AAA+ ring affects the ATPase activity. Disrupting the highly conserved D2915-R2976-D3042 salt-bridge network in the linker middle domain reduces the Rea1 ATPase activity by ≈50% (*n* = 3). The compromised ATPase activity indicates that the AAA+ ring engagement of the linker top and middle stimulates the Rea1 ATPase activity. Error bars show the standard deviation. **f** Three cryoEM structures of the Rea1_{D2915A-R2976A-D3042A} mutant in the presence of ATP. The linker conformation I is highly similar to the linker conformation of Rea1_{ΔAAA2H2α} ATPγS (compare Fig. 3a and Supplementary Fig. 8a, e). In conformations II and III the linker top and middle domains have swung towards the AAA+ ring (compare also Supplementary Fig. 18). Consistent with our previous domain assignments (compare Supplementary Movies 1 and 2), the region between the linker middle and stem domain acts as pivot point. Source data are provided as a Source Data file.

(Fig. 6d) and the correlated functional defects in pre60S particle maturation and nuclear export (Fig. 7). The fact that defects of the linker to correctly engage with the AAA+ ring also reduce the Rea1 ATPase activity (Fig. 6e) suggests that the interactions during linker - AAA+ ring engagement influence ATP hydrolysis. For example, such interactions might induce conformational changes in the Rea1 AAA+

ring that in turn stimulate the release of ATP hydrolysis products to boost ATPase activity.

ATP-hydrolysis in the AAA+ ring is required to engage the rotated linker top and middle domains with the AAA+ ring during states 6 – 8. The ultimate goal of the linker remodelling pathway is to bring the linker top into contact with the AAA+ ring-docked and substrate-

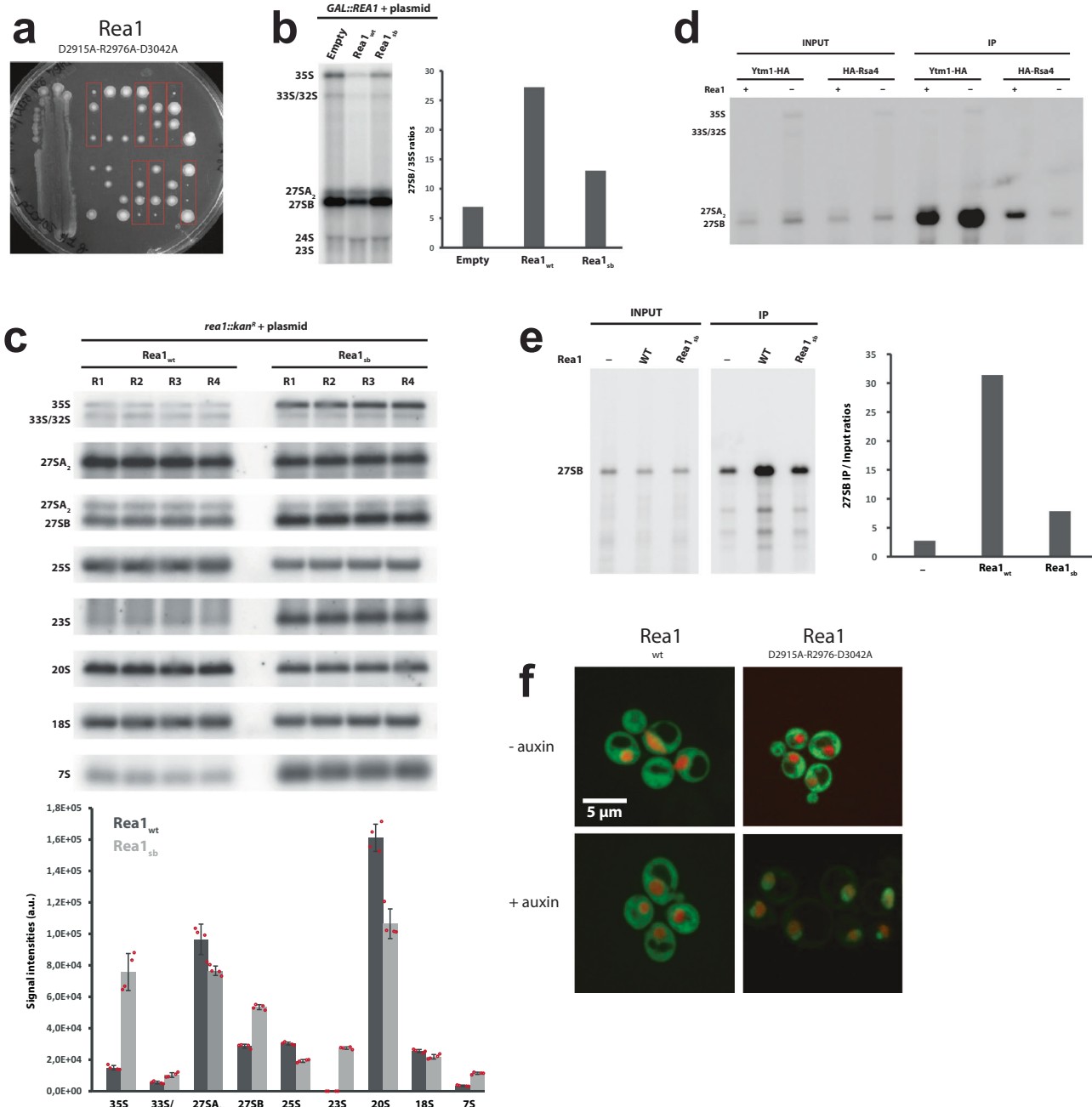

**Fig. 7 | Functional impact of mutations in the conserved salt-bridge network D2915-R2976-D3042 of the Rea1 middle domain. a** A heterozygous REA1/rea1::kanR diploid strain was transformed with a plasmid directing expression of the Rea1_{D2915A-R2976A-D3042A} mutant. After sporulation of the resulting strains, tetrads were dissected and haploid spores were spotted in rows on YPD medium. Spores containing the Rea1_{D2915A-R2976A-D3042A} mutant could support viability (red rectangles), although growth was strongly impaired. **b** Impact on pre-rRNA processing. Left panel: Northern analyses of pre-rRNA processing in GAL::rea1 cells expressing Rea1_{D2915A-R2976A-D3042A} ("Rea1_{sb}") or wild-type Rea1 (Rea1_{wt}) from plasmids or transformed with an empty vector (empty), shifted 24 hours on glucose-containing medium. Right panel: Quantification of 27SB/35S ratio. The Rea1_{D2915A-R2976A-D3042A} mutants displays reduced ratios compared to the Rea1_{wt} control indicative of pre60S maturation defects. Experiment was done once. **c** Upper panel: Detailed northern pre-rRNA processing analysis of rea1::kanR strain expressing Rea1_{D2915A-R2976A-D3042A} ("Rea1sb") from a plasmid. Four wt strains and four Rea1_{D2915A-R2976A-D3042A} expressing strains were analysed in parallel. Detection of the indicated (pre-)rRNAs by northern hybridization with anti-sense oligonucleotide probes. Lower panel: levels of the indicated (pre-)rRNAs in the Rea1_{D2915A-R2976A-D3042A} mutant or wild-type (n = 4). Error bars show the standard deviation. **d** Rea1 depletion affects the association of Rsa4 with pre60S particles. REA1 (Rea1 +) or GAL::rea1 (Rea1 -) strains expressing HA-tagged Ytm1 or Rsa4 were grown in glucose-containing medium. Immunoprecipitation experiments were then carried out with anti-HA agarose beads and the indicated pre-rRNAs in the input and immunoprecipitated (IP samples) were detected by northern analyses. The experiment was repeated twice. **e** Rea1_{D2915A-R2976A-D3042A} ("Rea1_{sb}") affects the association of Rsa4 with pre60S particles. Left panel: Analysis as in d, except that a GAL::rea1 strain expressing HA-tagged Rsa4 and Rea1_{D2915A-R2976A-D3042A} or wild-type Rea1 (Rea1_{wt}) from plasmids was used. -: strain transformed with an empty vector. Detection of 27SB pre-rRNA by northern hybridization. Right panel: ratios of the levels of precipitated 27SB pre-rRNA over input levels. Experiment was done once. **f** Disrupting the D2915-R2976-D3042 salt-bridge network leads to nuclear pre60S particle export defects. Experiments were repeated three times. Source data are provided as a Source Data file.

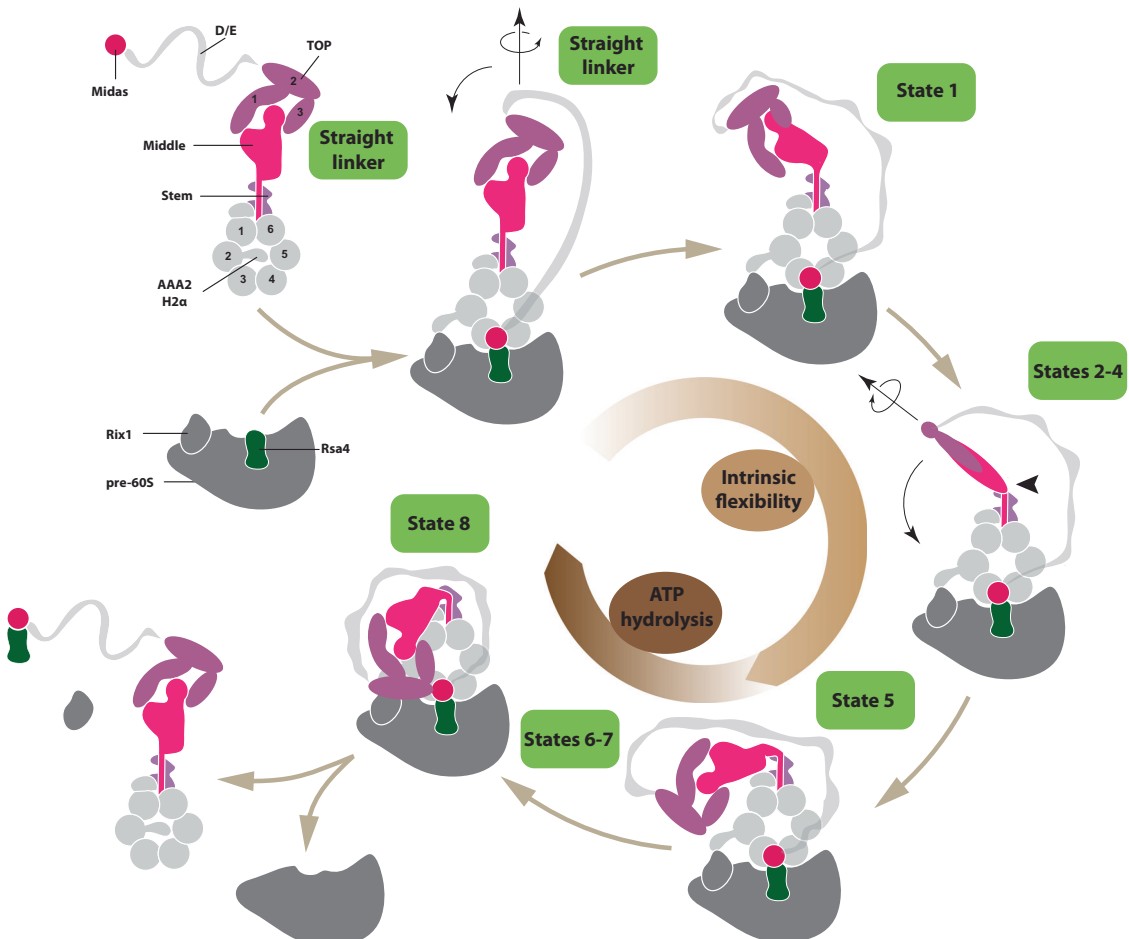

**Fig. 8 | Model for the Rea1 mechanism.** Rea1 in the absence of pre60S particles exists in an autoinhibited form with the AAA2H2α insert occupying the central pore of the AAA+ ring and the linker in the straight conformation. The binding of Rea1 to pre60S particles relocates AAA2H2α towards the pre60S particles, which allows the MIDAS domain to dock onto the AAA+ ring to engage with its assembly factor substrate (here Rsa4). The linker remains in the straight conformation. The linker subsequently rotates and pivots towards the plane of the AAA+ ring to reach state 1. From here the linker middle and top domains rotate around the long linker axis and swing towards the AAA+ ring. The region between the linker middle and stem domains acts as pivot point. In state 5, the linker middle and top domains are fully rotated and in close proximity to the AAA+ ring. In states 6-8, the linker top engages with the AAA+ ring and in the final remodelling step state 8 the linker top2/3 domains interact with the MIDAS domain to allow the transmission of force for assembly factor removal. Up to state 5, linker remodelling is nucleotide independent and driven by the intrinsic conformational flexibility of the linker. States 6–8 require ATP hydrolysis. Cartoon adapted from Sosnowski et al.[17] Elife 7, https://doi.org/10.7554/eLife.39163 (2018) under a CC BY license: https://creativecommons.org/licenses/by/4.0/.

engaged MIDAS domain. We have demonstrated that ATP hydrolysis driven Rea1 linker remodelling is able to produce mechanical force, which would allow the linker top to pull on the substrate engaged MIDAS domain to remove assembly factors from pre60S particles. The construct used in the microtubule gliding assays lacked large parts of the D/E- rich region and the complete MIDAS domain, which suggests that these structural elements are dispensable for force production. Since we used an ensemble assay to demonstrate force production, we cannot speculate on the magnitude of force produced by individual Rea1 molecules. Another limitation of these experiments is that we cannot estimate to which extend the mechanical force generated by linker remodelling is transmitted to the MIDAS domain for assembly factor removal. Optical trap experiments on individual Rea1 molecules will be required to answer these questions.

We consistently observed that the Rea1 linker states associated with ATP-hydrolysis are sampled only by a minority of particles in our data sets. One possible explanation could be that these states are simply transient and extremely short-lived during the mechanochemical cycle of Rea1. Another possibility is that additional factors are required to further stabilize them such as protein/RNA interaction partners provided/present in the context of pre60S particle binding.

Since we investigated Rea1 in the absence of pre60S particles, a potential caveat of this study is that linker remodelling might be different in the presence of pre60S particles. The interaction between Rea1 and the pre60S environment might influence the kinetics and lifetime of linker remodelling sates. It might also induce additional remodelling steps. Since Rea1 interacts with two different pre60S particles, a nucleolar in the case of Ytm1, and a nucleoplasmic in the case of Rsa4, one could also imagine the occurrence of additional/modified linker remodelling steps depending on the pre60S context. To ultimately answer such questions, different Rea1 linker remodelling states will have to be analysed by high-resolution cryoEM structures of Rea1-pre60S particle complexes.

We have focused our analysis on the linker remodelling states that consistently appeared in comparable negative stain EM data sets. States 1–5 suggest that the swing of the linker towards the AAA+ ring and the rotation of the linker top are correlated events, i.e. the linker rotates as it swings towards the AAA+ ring. However, this correlation is not completely strict as in some of the data sets additional linker extended and intermediate states appeared, suggesting that the linker can also swing without rotation or fully rotate before reaching the proximity of the AAA+ ring (Supplementary Fig. 21). The occurrence of

these states indicates that alternative pathways exist to remodel the linker from state 1 to state 5.

It is not clear why Rea1 would follow such an elaborate scheme to remove assembly factors from pre60S particles. One possibility is that states 1–5 are part of a sensing mechanism. After Rea1 has bound to pre60S particles, the intrinsic conformational flexibility represented by these states would allow the linker top to repeatedly come close to the AAA+ ring. It is conceivable that the pre60S environment influences the ATPase activity of the Rea1 AAA+ ring. In case essential maturation events have not occurred yet, negative stimuli could inhibit the ATPase activity, which would prevent the linker from engaging with the AAA+ ring. The successful completion of maturation steps could stimulate the Rea1 ATPase activity and allow the linker to engage with the AAA+ ring and ultimately the MIDAS domain to remove assembly factors from pre60S particles.

Currently, there are two mechanistic models for Rea1 functionality. Early structural studies on Rea1 favoured a linker remodelling hypothesis to remove assembly factors from pre60S particles[10], whereas more recently an alternative AAA+ ring model was suggested[20,23]. While both models acknowledge that ATP hydrolysis in the AAA+ ring is required for assembly factor removal, they differ with respect to the question how ATP hydrolysis driven conformational changes in the AAA+ ring catalyse assembly factor removal. In the AAA+ ring model, such conformational changes are directly communicated to the AAA+ ring docked and substrate-engaged MIDAS domain to produce force for pre60S assembly factor removal. This model attributes a more passive role to the Rea1 linker suggesting it acts as a "fishing post" for the flexibly attached MIDAS domain and is not involved in force production for assembly factor removal[20]. The linker remodelling hypothesis states that ATP-hydrolysis in the AAA+ ring drives the remodelling of the Rea1 linker, which in turn provides the force for assembly factor removal from pre60S particles. In this model, ATP-hydrolysis driven conformational changes in the AAA+ ring indirectly catalyse assembly factor removal from pre60S particles via linker remodelling. The results presented here favour the linker remodelling hypothesis. Especially the data on the Rea1$_{D2915A-R2976A-D3042A}$ mutant suggest that a potential AAA+ ring based force production for assembly factor removal is not essential to the Rea1 mechanism. Our structural characterization of these mutants clearly indicates that AAA+ ring and linker are well folded, but they nevertheless show severe functional defects. While these defects can be explained by the linker remodelling hypothesis, the AAA+ ring model would expect these mutants to be functional. However, it is still possible that AAA+ ring based force production plays a supportive role in linker driven assembly factor removal.

Together with the motor protein dynein and the E3 ubiquitin ligase RNF213, Rea1 forms a special subclass within the AAA+ field. All these proteins have their six AAA+ modules concatenated into a single gene, whereas the hexameric AAA+ rings of the vast majority of AAA+ machines are formed by association of individual protomers. In the case of dynein and Rea1, ATP-hydrolysis in the AAA+ ring drives the remodelling of an AAA+ ring linker extension, which in both cases correlates with the production of mechanical force. With around 400 amino-acid residues, the dynein linker is much smaller than the Rea1 linker, which spans over around 1700 amino-acid residues, and there are no structural similarities between them. The dynein linker switches between a straight post-powerstroke and bent pre-powerstroke conformation[26] illustrating that the structural heterogeneity is much smaller compared to the Rea1 linker, which is able to adopt at least 9 different conformations (straight linker conformations + states 1–8). These differences might be explained by the purpose of force production in dynein and Rea1. The processive movement along microtubules requires highly repetitive linker remodelling. In this context, the switch between two remodelling states ensures efficient and fast force production. Rea1 on the other hand functions as a quality control machine. The main task is not highly repetitive force production but rather a single force production event at a precisely controlled time point. In this context, many linker remodelling steps offer the opportunity to integrate various signals from the pre60S environment to trigger assembly factor removal only when essential pre60S maturation events have occurred. The linker of the E3 ubiquitin ligase RNF213 is similar to the straight post-powerstroke dynein linker[27]. However, so far no alternative linker remodelling states have been reported for RNF213. It has recently been demonstrated that ATP binding to the AAA+ ring stimulates the ubiquitin ligase activity[28], but it is not clear if this activity increase correlates with RNF213 linker remodelling.

This study highlighted the general mechanistic principles of Rea1. Fully understanding the molecular basis of the Rea1 mechanism will require high resolution investigations on the various linker remodelling states. Especially the AAA+ ring engaged states will be of great interest as these are the ones associated with ATP-hydrolysis and mechanical force production. The extreme low abundance of these states and the structural flexibility of the linker will make such investigations challenging.

## Methods

### Molecular cloning, protein expression and purification

The Gibson assembly approach was used for all molecular cloning. The sequences of all primers use are provided in Supplementary Table 3. The centromeric plasmids used for the *S. cerevisiae* pre60S export assay, tetrad dissection as well as the pre-rRNA processing analysis harboured a *URA3* selection marker and the endogenous *rea1* promotor and terminator regions. To generate the centromeric plasmid with the endogenous Rea1 promotor and terminator, we use plasmid pRS416 as template. Primers GK1 and Gk2 were used to linearize pRS416. Primers GK3 and GK4 as well as GK5 and GK6 were used to amplify the Rea1 promotor and terminator regions from yeast genomic DNA, respectively.

In order to generate the centromeric *Rea1$_{wt}$* plasmid, the plasmid described above was linearized using primes GK7 and GK8 to clone *Rea1$_{wt}$* or mutant sequences in-between the Rea1 promotor and terminator regions. The *Rea1$_{wt}$* sequence was amplified from a previously described expression plasmid[17] using primers GK9 and GK10. The *Rea1Δ$_{middle-top}$* construct was generated using the centromeric *Rea1$_{wt}$* plasmid as a template. Primers GK11 and GK12 were used to linearize the plasmid omitting the linker middle and top domains. Primers GK13 and GK14 were used to amplify the α-helical extension of the middle domain and to fuse it in-between the linker stem and the D/E rich region with GG and GSGS spacers, respectively. (Rea1Δ$_{middle-top}$: M1-S2608 + GG + D2911-D2967 + GSGS + F4049-S4910). The alanine mutations for the Rea1$_{D2915A-R2976A-D3042A}$ salt-bridge network mutant were introduced in the centromeric *Rea1$_{wt}$* plasmid with primers GK15-GK20.

The recombinant expression and purification of *S. cerevisae* Rea1$_{wt}$, Rea1$_{ΔAAA2H2α}$ and Rea1$_{ΔAAA2H2α+ΔMIDAS}$ were done as recently described[17]. Briefly, the constructs were cloned into a pYES2 vector harbouring a Pgal promoter followed by an N-terminal tandem Protein-A tag followed by two preScission protease cleavage sites. For the construct used in the microtubule gliding assays, we anticipated that the flexible, unstructured and highly negatively charged D/E rich region (theoretical pI: 3.77) of Rea1 might prevent the binding of microtubules. Therefore, we truncated ≈80% of the D/E-rich region as well as the whole MIDAS domain using primers GK21 and GK22 with the *Rea1$_{ΔAAA2H2α}$* expression plasmid as template. The resulting construct, Rea1$_{ΔAAA2H2α+Δ4168-4907}$, was further modified by fusing the spytag[29] with two flanking spacer glycines (GAHIVMVDAYKPTKG) between E3943 and T3944 in the linker top2 domain using primers GK23 and GK4. The GFP sequence was amplified from the *Rea1$_{wt}$* expression plasmid using primers GK25 and GK26 and introduced with two flanking spacer glycines in the AAA5S domain between S1902 and

I1903 with primers GK27 and GK28. To generate the expression plasmid for the Rea1$_{\Delta AAA2H2\alpha}$ Walker-B mutant, the Walker-B glutamates in the four active AAA2-AAA5 sites were mutated to glutamine (E830Q, E1151Q, E1493Q, E1816Q) using PCR mutagenesis with primers GK29-GK37 and the *Rea1$_{\Delta AAA2H2\alpha}$* expression plasmid as template. The expression plasmid for the Rea1$_{D2915A-R2976A-D3042A}$ mutant was constructed using PCR mutagenesis with primers GK15-GK20 and the Rea1$_{wt}$ expression plasmid[17] as a template.

In order to express the constructs, corresponding plasmids were transformed into *S. cerevisiae* strain JD1370 using the *URA3* selection marker and expression of the constructs was induced by shifting cells to galactose containing media. Cells were pelleted, resuspended in water, flash frozen in liquid nitrogen and stored at −80 °C. Frozen pellets were blended and further purified by IgG sepharose affinity and gel filtration chromatography as described in ref. [17] with the exception that 50 mM Hepes pH 7.5, 150 mM NaCl, 2 mM MgCl$_2$ and 1 mM DTT was used as gel filtration buffer. For the EDTA purified Rea1 wt sample (Supplementary Fig. 2), the gel filtration buffer was modified to 50 mM Hepes pH 7.5, 150 mM NaCl, 5 mM EDTA and 1 mM DTT.

The spycatcher-dynein MTBD construct was generated by fusing the C-terminus of a chimeric seryl-tRNA synthetase-22:19 dynein MTBD construct in the high microtubule affinity state[30] to the N-terminus of the spycatcher domain[29] in a pET42a vector background using primers GK38/GK39 and GK40/GK41, respectively. The construct was expressed in *Escherichia coli* strain BL21 (DE3) by induction with 1 mM isopropyl β-D-thiogalactopyranoside for 16 h at 18 °C. All subsequent steps were carried out at 4 °C. The cells were harvested by centrifugation at 5000 g for 15 min. The pellet was resuspended in lysis buffer (50 Hepes pH 7.5, 200 mM NaCl, 10% glycerol, 1 mM 2-mercaptoethanol, 5 mM imidazole, 0.5 mM phenylmethylsulfonyl fluoride) and lysed using a sonicator. The lysate was subsequently clarified by centrifugation at 80000 g for 30 min. The expressed protein was incubated with nickel-nitrilo- triacetic acid beads for 1 h and washed with three column volumes of wash buffer (50 mM Hepes pH 7.5, 200 mM NaCl, 10% glycerol, 1 mM 2-mercaptoethanol, 30 mM imidazole, 1 mM Mg-ATP). The protein was eluted with elution buffer (50 mM Hepes pH 7.5, 200 mM NaCl, 10% glycerol, 1 mM 2-mercaptoethanol, 250 mM imidazole).

The full-length *C. thermophilum* Rea1 MIDAS domain (H4690-S4997) was amplified from a cDNA library using primers TK1 and TK2 and cloned in a pETN3 vector backbone with an N-terminal 2x strep tag using primers TK3 and TK4. The MIDAS$_{\Delta 4734-P4775}$ construct was generated by replacing the P4734-P4775 loop (equivalent to the conserved E4656-K4700 loop in the S. cerevisiae MIDAS domain) with a GSGSG linker using primers DV1 and DV2. The *C. thermophilum* Rea1 linker top2/top3 construct (G3636-G4113) was amplified from a cDNA library using primers TK5 and TK6 and cloned in a pETN4 vector with an N-terminal GST tag using primers TK7 and TK8. The MIDAS plasmids were transformed into *E. coli* BL21 pLysS and the linker top2/top3 construct into *E. coli* BL21 pRIL. The cells were grown in LB (*E. coli* BL21 pLysS) or 2x TB (*E. coli* BL21 pRIL) media in the presence of the appropriate antibiotics (AMP + CHLOR) at 37 °C to an OD$_{600}$ of 0.6. To induce expression, the temperature was adjusted to 18 °C and 1 mM IPTG was added to the medium. The cells were harvested after 18 h by centrifugation at 4500 g for 15 min. The pellets were resuspended in lysis buffer (50 mM Hepes pH 7.5, 150 mM NaCl, 2 mM MgCl$_2$, 0.2% v/v NP-40 and 10% v/v glycerol) and Roche protease inhibitor was added. Cells were lysed by passing them 3x through an Emulsoflex at 1500 psi. The lysate was centrifuged at 56000 g for 30 minutes. The supernatant was incubated with either Streptactin- (MIDAS constructs) or GST-Speharose (linker top2/top3) beads for 1 h at 4 °C. The beads were washed with wash buffer (50 mM Hepes pH 7.5, 150 mM NaCl, 2 mM MgCl2 and 10% v/v glycerol). The proteins were eluted with wash buffer + 50 mM biotin (MIDAS) or wash buffer + 75 mM GSH (linker top2/top3). The MIDAS constructs were further purified by running

them over a superdex 200 column in gel filtration buffer (50 mM Hepes pH 7.5, 150 mM NaCl and 2 mM MgCl2). The linker top2/top3 construct was dialysed against wash buffer O/N to remove the GSH.

For the yeast two-hybrid assays, the sequences encoding the MIDAS domain (E4623-S4910), the linker top2/top3 (Y3557-N4041) the shorter top2/top3 domain (I3601-N4041) or just the top3 region (D3786-E3905) were inserted into the NdeI and BamHI sites of pGADT7 AD and pGBKT7 vectors (Clontech) using the In-Fusion Snap Assembly Cloning protocol (Takara Bio USA, Inc.). The following primer pairs were used to amplify the different Rea1 domains: MIDAS in pGADT7 AD: OHA631 and OHA632; top2/top3 in pGADT7 AD: OHA633 and OHA634; top2/top3 short in pGADT7 AD: OHA635 and OHA636; top3 in pGADT7 AD: OHA637 and OHA638; MIDAS in pGBKT7: OHA639 and OHA640; top2/top3 in pGBKT7: OHA641 and OHA642; top2/top3 short in pGBKT7: OHA643 and OHA644; top3 in pGBKT7: OHA645 and OHA646.

## Yeast strains and media

For the pre60S export assay, an auxotrophic *S. cerevisiae* strain (-*URA*, -*LEU*, -*HIS*, -*TRP*) featuring an mCherry fused to the C-terminus of the endogenous histone 2B locus was used. The strain was further modified by tagging the C-terminus of the endogenous Rea1 with the IAA17 protein using kanamycin selection and the IAA17 partner protein OsTir1 was integrated into the H0 locus using hygromycin selection. The addition of auxin to the medium triggers the IAA17-OsTir1 interaction and subsequent proteasome degradation[31]. This strain was used to transform a pRS415 plasmid harbouring Rpl25-GFP using the -*LEU2* selection marker. The strain was further transformed with the centromeric plasmids harbouring *Rea1$_{wt}$*, *Rea1$_{\Delta middle-top}$* and *Rea1$_{D2915A-R2976A-D3042A}$* using -*LEU2* -*URA3* double selection.

The diploid REA1/rea1::kan$^R$ strain in the BMA64 genetic background (*MATa/MATα, MATa; his3-11_15/his3-11_15; leu2-3_112/leu2-3_112; ura3-1/ura3-1; trp1Δ2/trp1Δ2; ade2-1/ade2-1; can1-100/can1-100*) used for the tetrad dissection assays was constructed as follows. The rea1::kan$^R$ deletion cassette was mobilized by PCR amplification of genomic DNA extracted from the commercial REA1/rea1::kan$^R$ diploid strain in the BY4741 background (purchased from Open Biosystems), using primers OHA532 and OHA534.

The GAL::REA1 strain was obtained by transforming haploid BY4741 strain (*MATa, his3Δ1, leu2Δ0, met15Δ0, ura3Δ0*) with a kan$^R$-PGAL1 cassette flanked by REA1 sequences corresponding to the promoter region and the beginning of the open reading frame at the 5' and 3' ends, respectively. This cassette was produced by PCR amplification of plasmid pFA6a-kanMX6-PGAL1[32] using oligonucleotides OHA529 and OHA530. Genomic insertion of the cassette has been verified by PCR amplification of genomic DNA purified from selected transformants and oligonucleotides OHA532 and OHA533.

YTM1-HA and GAL::REA1/YTM1-HA strains were obtained by transforming the BY4741 or the GAL::REA1 strains with a 3HA-His3MX6 PCR cassette flanked by YTM1 open reading frame and terminator sequences at the 5' and 3' ends, respectively, produced by amplification of plasmid pFA6a-3HA-His3MX6[32] with oligonucleotides OHA543 and OHA544. Genomic insertion of the cassette has been verified by PCR amplification of genomic DNA purified from selected transformants and oligonucleotides OHA545 and OHA546.

HA-RSA4 and GAL::REA1/HA-RSA4 strains were obtained by CRISPR-Cas9-mediated genome editing of the BY4741 or the GAL::REA1 strains using the procedure described in[33]. Oligonucleotides OHA553 and OHA554 were annealed and cloned into pML104 vector to generate a plasmid expressing the Cas9 nuclease and a guide RNA targeting the nuclease in the vicinity of the ATG of RSA4 open reading frame. This plasmid was transformed into the BY4741 or the GAL::REA1 strains along with a repair PCR fragment consisting in the 3HA tag sequence flanked by RSA4 sequences corresponding to the promoter region and the beginning of the open reading frame at the 5' and 3' ends, respectively. This cassette was produced by PCR

amplification of plasmid pFA6a-3HA-kanMX6[32] with oligonucleotides OHA551 and OHA552. Transformants were selected on synthetic medium (see below) lacking uracil to select clones having internalized the recombinant pML104 plasmid. The plasmid was then rapidly counter-selected on synthetic YNB medium containing 5-Fluoroorotic acid. Genomic insertion of the cassette in the resulting clones has been verified by PCR amplification of genomic DNA purified from selected transformants and oligonucleotides OHA559 and OHA560.

Transformed REA1/rea1::kanR diploid strains were sporulated on 2% potassium acetate, 0.22% yeast extract, 0.05% glucose, 0.079% complete supplement mixture (MP biomedicals), 2% agar, adjusted to pH 7.0 with potassium hydroxide. Tetrads were dissected using a Singer MSM dissection microscope.

Strains were grown either in YP medium (1% yeast extract, 1% peptone) (Becton-Dickinson) supplemented with 2% glucose or 2% galactose as the carbon source or in synthetic medium (0.17 % yeast nitrogen base (MP Biomedicals), 0.5% $(NH_4)_2SO_4$) supplemented with 2% glucose or 2% galactose and the required amino acids. Selection of the kanamycin-resistant transformants was done by addition of G418 to a final concentration of 0.2 mg/ml.

### *S. cerevisiae* pre60S export assays

Strains carrying the Rpl25-GFP and Rea1 construct plasmids were used to inoculate a 50 ml pre-culture of CSM -leu -ura minimal medium enriched with 2% glucose and 0.1 mM adenine. The pre-culture was incubated over night at 200 rpm and 30 °C. The following morning the preculture was used to inoculate another 50 ml of CSM -leu -ura minimal medium enriched with 2% glucose and 0.1 mM adenine to a final OD of 0.2. In order to induce the degradation of endogenous Rea1, auxin to a final concentration of 0.1 mM was added. The culture was incubated at 200 rpm and 30 °C and cells were imaged after 6 h.

### Negative stain electron microscopy

Standard plain carbon grids (copper 300 mesh, vendor: EMS) were glow discharged in air plasma (30 s, -2.5 mA, 0.21 mbar). Rea1 samples were diluted in 50 mM Hepes 7.5, 150 mM NaCl, 2 mM $MgCl_2$ and 1 mM DTT to a final concentration of 25 nM in the presence of 3 mM nucleotide or without nucleotide.

5 µl of the 25 nM sample were applied to the glow discharged grid for 2 min. After side blotting out the excess of sample, grids were stained following standard procedures with freshly centrifuged and filtered 2% Uranium Acetate.

Grids were imaged using a FEI Technai G2 FEG TEM operated at 200 kV HT equipped with a charge-coupled device camera (2 K US10001; Gatan). Acquisitions were made with SerialEM (version 3-7-14)[34] at 50kx magnification(2.002 A/pix), a nominal defocus of −1.5 µm and a 5 × 5 montage mode for speeding up data collection. Micrographs were extracted from montage stacks using IMOD[35] and processed using Relion (version 3.1)[36].

The Rea1$_{\Delta AAAA2H2\alpha}$ ATPγS data set (Supplementary Fig. 22) consisted of a total of 74141 particles, which was sufficient to resolve all 8 linker remodelling states. The Rea1$_{wt}$ ATP, APO, AMPPNP and ATPγS data sets had 70952, 69908, 76407 and 87826 particles. The Rea1$_{\Delta AAAA2H2\alpha}$ ATP, ADP, AMPPNP, ATPγS + PhoX and APO data sets consisted of 125294, 97479, 387291, 85123 and 314443 particles. The data set for the Rea1$_{\Delta AAAA2H2\alpha}$ Walker-B mutant in the presence of ATP consisted of 133585 particles. The Rea1$_{D2915A-R2976A-D3042A}$ ATP data sets consisted of 69331 particles. The Rea1$_{\Delta AAAA2H2\alpha}$ ATPγS data set was further processed by initial model generation, 3D classification and 3D refinement in Relion (version 3.1)[36]. The final reconstruction consisted of 33244 particles and refinement to a nominal resolution of 17.02 Angstrom.

### CryoEM electron microscopy

For the Rea1$_{\Delta AAAA2H2\alpha}$ ATPγS and Rea1$_{D2915A-R2976A-D3042A}$ ATP data sets, standard copper/rhodium quantifoil R2/2 with an additional carbon layer and streptavidin affinity grids[37] were used. In addition, standard copper/rhodium quantifoil R2/2 without additional carbon layer were used for Rea1$_{D2915A-R2976A-D3042A}$ ATP. 3 µl of Rea1$_{\Delta AAAA2H2\alpha}$ or Rea1$_{D2915A-R2976A-D3042A}$ at a concentration of 150 nM in 50 mM Hepes 7.5, 150 mM NaCl, 2 mM $MgCl_2$, 1 mM DTT, 0.005% DDM and 3 mM ATPγS or ATP were incubated with plasma cleaned (80% oxygen / 20% argon, 30% power, 90 sec) carbon grids. In case of the streptavidin affinity grids, 3 µl of biotinylated Rea1$_{\Delta AAAA2H2\alpha}$ or Rea1$_{D2915A-R2976A-D3042A}$ at a concentration of 60 nM were used. The streptavidin affinity grids were further treated as described recently[37]. For the unsupported Rea1$_{D2915A-R2976A-D3042A}$ ATP grids, 3 µl at a concentration of 1800 nM in the buffer described above were used. All grids were plunge frozen with a Vitrobot Mark IV robot (FEI), maintained at 95% humidity and 10 °C.

All data sets were collected on a FEI Titan Krios operating at 300 kV equipped with a Cs corrector and a Gatan K3-Summit detector using a slit width of 20 eV on a GIF-quantum energy filter (Gatan). In case of the carbon supported grids, two data sets where acquired at 0° and 30° tilt. All Krios datasets were acquired at 81kx magnification (0.862 A/pix). Acquisitions were performed semi-automatically, managed with Serial EM[34].

Micrographs were processed with relion (version 3.1)[36]. The streptavidin crystal pattern in the micrographs of the streptavidin affinity grids was erased in Fourier space using python scripts described in ref. 38 (https://github.com/NilsMarechal/SAGsub; the script is also available directly from Nils Marechal, marechan@igbmc.fr). Particles were extracted with a binning of 4 times. The combined particles datasets were submitted to 2D classifications, stochastic gradient decent to create initial models and several rounds of 3D classification from which the straight linker conformation emerged as the only interpretable cryoEM map in the case of the Rea1$_{\Delta AAAA2H2\alpha}$ ATPγS data set. For the Rea1$_{D2915A-R2976A-D3042A}$ ATP data set, three linker conformations (I-III) could be obtained with conformation I being highly similar to the straight linker conformation of Rea1$_{\Delta AAAA2H2\alpha}$ ATPγS cryoEM map. The Rea1$_{\Delta AAAA2H2\alpha}$ ATPγS data set had 230k particles. The final map of the straight linker as shown in Fig. 3a consisted of 100k particles and refined to Nyquist spacing (7.1 Å for a pixel size of 3.45 Å/pixel). The Rea1$_{D2915A-R2976A-D3042A}$ data set had 1035k particles. The final models for Rea1$_{D2915A-R2976A-D3042A}$ ATP conformations I – III consisted of 320k, 250k and 240k particles and also refined to nomial Nyquist spacing ($\approx 7$ Å for a pixel size 3.45 Å/pixel). The Rea1$_{\Delta AAAA2H2\alpha}$ ATPγS and Rea1$_{D2915A-R2976A-D3042A}$ ATP (conformation I, II, III) maps have been submitted to the electron microscopy data bank with access codes EMD−50815 [https://www.ebi.ac.uk/emdb/EMD−50815], EMB-50816 [https://www.ebi.ac.uk/emdb/EMD−50816], EMD-50817 [https://www.ebi.ac.uk/emdb/EMD−50817] and EMD-50818.

### Crosslinking mass spectrometry

**Sample preparation and PhoX crosslinking.** Rea1$_{\Delta AAAA2H2\alpha}$ in 50 mM Hepes, 150 mM NaCl, 2 mM MgCl2 was incubated with 5 mM ATPγS, 5 mM AMP-PNP or left in the absence of nucleotide (APO). The final protein concentration after nucleotide incubation was 1.25 mg/ml. A 2 mg aliquot of PhoX crosslinker (Disuccinimidyl Phenyl Phosphonic Acid, Bruker)[39] was freshly diluted in DMSO to a final concentration of 5.16 mM. Each Rea1$_{\Delta AAAA2H2\alpha}$ sample was split into three aliquots (3x ATPγS, 3x AMPPNP, 3x APO) of 15 µg protein each and subsequently incubated with 1 µL of PhoX crosslinker stock solution, corresponding to a 200 molar excess of PhoX. The crosslinking reaction was carried out at 20 °C for 45 min and quenched by adding Tris-HCl to a final concentration of 10 mM and an additional 20 min incubation step. A fourth 15 µg ATPγS aliquot was crosslinked and kept for negative-stain electron microscopy analysis (Supplementary Fig. 13B). To control the reaction efficiency, 1.4 µg of the non-crosslinker and crosslinked samples were kept for a mass photometry analysis to check for the

appearance of non-specifically crosslinked dimers (Supplementary Fig. 13A).

After the quality check, all samples were reduced by adding DTT to a final concentration of 5 mM and incubation at 37 °C for 30 min. The subsequent alkylation was carried out by adding Iodoacetamide to a final concentration of 15 mM followed by an 1 hour incubation step in the dark. The samples were further processed by digesting them overnight with a Trypsin/Lys-C mix (Promega, Madison, USA) at a 50:1 substrate:enzyme ratio (w/w) at 37 °C overnight. The digestions were finally quenched with 1% TFA.

**Automated peptide cleanup and enrichment of PhoX-crosslinked peptides.** Peptides were first cleaned up by using the AssayMAP Bravo platform (Agilent Technologies; Santa Clara, California) with 5 µL C18 cartridges (Agilent). Cartridges were primed with 100 µl 0.1% TFA in 80% ACN and equilibrated with 50 µl 0.1% TFA in H$_2$O. 180 µl of digested peptides diluted in equilibration buffer were loaded on the cartridges and washed with 50 µl equilibration buffer. Peptides were eluted with 50 µl 0.1% TFA in 80% ACN. In order to enrich the PhoX-crosslinked peptides, 5 µl Fe(III)-NTA cartridges (Agilent) were primed with 100 µL of 0.1% TFA in H$_2$O and equilibrated with 50 µl 0.1% TFA in 80% ACN[39]. 140 µl of the C18-eluted peptides diluted in 0.1% TFA in 80% ACN were loaded and subsequently washed with 50 µl 0.1% TFA in 80% ACN. PhoX-crosslinked peptides were eluted with 50 µl of 1% NH$_4$OH solution and stored at -80 °C prior to the mass spectrometry analysis.

**LC-MS/MS acquisition and data processing.** Before being injected, PhoX-crosslinked peptides were dried in a SpeedVac concentrator and resuspended in 10 µl of 2% ACN/0.1% formic acid. NanoLC-MS/MS analysis was performed using a nanoAcquity Ultra-Performance-LC (Waters, Milford, USA) hyphenated to a Q-Exactive Plus Orbitrap mass spectrometer (Thermo Fisher Scientific, Bremen, Germany) equipped with a nanoSpray source. Samples were first trapped on a nanoACQUITY UPLC precolumn (C18, 180 µm × 20 mm, 5 µm particle size), prior to separation on a nanoACQUITY UPLC BEH130 column (C18, 75 µm × 250 mm with 1.7 µm particle size, Waters, Milford, USA) maintained at 60 °C. A Gradient of 105 min was applied using mobile phases A (0.1% v/v formic acid in H$_2$O) and B (0.1% v/v formic acid in ACN). The following conditions were applied: 1–3 % B for 2 min, 3–35% B for 77 min, 35–90% B for 1 min, 90% B for 5 min, 90–1% B for 2 min and finally 1% B maintained for 2 min (flow rate of 400 nl/min). The Q Exactive Plus Orbitrap source temperature was set to 250 °C and the spray voltage to 1.8 kV. Full scan MS spectra (300–1800 m/z) were acquired in positive mode (resolution of 140,000, max. injection time of 50 ms, AGC target value of 3.106) with the lock-mass option enabled (polysiloxane ion from ambient air at 445.12 m/z). Using a Data Dependant Acquisition strategy, the 10 most intense peptides per full scan (charge states > 2) were isolated using a 2 m/z window and fragmented using stepped collision energy HCD (27%, 30%, 33% normalized collision energy). MS/MS spectra were acquired with a resolution of 35 000, a maximum injection time of 100 ms, an AGC target value of 1.105, and a dynamic exclusion time of 60 s. NanoLC-MS/MS system was piloted with XCalibur software v3.0.63, 2013 (Thermo Scientific) and a NanoACQUITY UPLC console v1.51.3347 (Waters).

Raw data were directly processed with Thermo Proteome Discoverer 2.5.0.400 (Thermo Scientific) using the XlinkX[40] node for identification of crosslinks and the Sequest HT node for the identification of linear peptides. For both linear and crosslinked peptides searches, Cystein carbamidomethylation was set as fixed modification. Methionine oxidation, N-term acetylation, tris-quenched monolinks and water-quenched monolinks were set as dynamic modifications. Trypsin was set as the cleavage enzymes with minimal length of 7 amino acids, 2 (linear peptides) and 3 (crosslinked peptides) missed cleavages were allowed. To increase confidence, identification were only accepted for a minimal score of 40 and a minimal delta score of

4[39]. A 1% false discovery rate was also applied at a crosslinked peptides level (XlinkX validator node). In-house database for linear peptides identification was composed of 187 entries (Rea1$_{\Delta AAA2H2\alpha}$, common contaminants and reversed sequences), database for crosslinks identification was only composed of the Rea1$_{\Delta AAA2H2\alpha}$ sequence (purified sample) to reduce the search space. For the consensus step, proteins identifications were controlled at a 1% FDR in the protein validator node. Out of the three replicates performed, cross-linking interactions were validated when seen in at least 2 out of 3 replicates. Validated (file threshold 2/3) unique cross-links were visualized on Rea1 sequence and plotted on the different PDB structures using xiVIEW webserver (www.xiview.org). Both XiView server and PyMol Molecular Graphics System (version 2.5.4, Schrödinger, LLC) were used to visualize and measure XLs Cα-Cα distances on the structure. Corresponding distances were only validated if within ≤ 25 Å threshold. The XL-MS dataset of Rea1 cross-linked with 200 molar excesses of PhoX generated in this study (apo state, in presence of AMP-PNP or in presence of ATPgS), including experimental settings and XL identification results has been deposited to the ProteomeXchange Consortium via the PRIDE[41] partner repository with the dataset identifier PXD053636.

**Mass Photometry.** Mass Photometry (TWOMP, Refeyn Ltd, Oxford, UK) was performed on the crosslinked apo, ATPγS and AMPPNP Rea1$_{\Delta AAA2H2\alpha}$ replicates as well as on the ATPγS Rea1$_{\Delta AAA2H2\alpha}$ non-crosslinked control (Supplementary Fig. 13A). Microscope cover slides (24 × 50 mm, 170 ± 5 µm, No. 1.5H, Paul Marienfeld GmbH & Co. KG, Germany) were cleaned with milli-Q water, isopropanol, milli-Q water and were then dried with a clean nitrogen stream[42]. Six-well reusable silicone gaskets (CultureWellTM, 50 – 3 mm DIA × 1 mm Depth, 3-10 µL, Grace Bio-Labs, Inc., Oregon, USA) were carefully cut and assembled on the cover slide center. After being placed in the mass photometer and before each acquisition, an 18 µL droplet of Phosphate Buffer Saline (PBS) was put in a well to enable focusing on the glass surface. All samples were first diluted with their native buffer, and then 2 µL of the protein stock solution was diluted into the 18 µL PBS droplet. A contrast-to-mass calibration was performed with Bovine Serum Albumin, Bevacizumab, and L-Glutamate Dehydrogenase diluted in PBS buffer, pH 7. For analysis, samples were diluted just below the saturation to obtain the highest number of counts without losing signal quality (typically between 10 to 40 nM in the droplet). Three movies of 60 s were recorded for each native ATPγS sample and two for each XL sample with AcquireMP, and processed with DiscoverMP softwares (Refeyn Ltd, Oxford, UK).

**GST-linker top2/3 – MIDAS domain pulldown**
*C. thermophilum* GST-linker top2/3 construct or GST-8xHis-2x protein A-2xprecission protease site construct (control) and the full length *C. thermophilum* MIDAS domain or the MIDAS$_{\Delta 4734-P4775}$ construct were mixed at a molar ratio of 1: 10 in 30 mM Hepes pH 7.5, 100 mM NaCl and 2 mM MgCl$_2$ and incubated on ice for 1 h. Glutathione sepharose (Cytiva) was added and the samples were further incubated for 1 h on ice. The samples were centrifuged at 400 g and 4 °C for 2 min and the supernatant was carefully removed without disrupting the Glutathione sepharose pellet. The Glutathione sepharose pellet was subsequently washed with 20 mM Hepes pH 7.5, 50 mM NaCl and 2 mM MgCl$_2$. The centrifugation/washing step was repeated 5x for all samples. GSH to final concentration of 25 mM was added and all samples were incubated at RT for 30 min. After a final round of centrifugation at 400 g for 5 min, the supernatant of all samples was analysed by SDS-PAGE and Coomassie blue or silver staining.

**Microtubule gliding assays**
90 µl of Rea1$_{\Delta AAA2H2\alpha+\Delta 4168-4907}$ (0.7 mg/ml) were mixed 60 µl of spycatcher-dynein MTBD (3.8 mg/ml) and 150 µl 200 mM NaOH-PIPES pH 6.8 and incubated for 4 h at room temperature to catalyze the

formation of the covalent spycatcher-spytag bond. The reaction mixture was subsequently run over a superpose 6 gel filtration column to separate the $\text{Rea1}_{\Delta AAA2H2\alpha+\Delta4168-4907}$ -spycatcher-dynein MTBD construct from the unreacted $\text{Rea1}_{\Delta AAA2H2\alpha+\Delta4168-4907}$ and spycatcher-dynein MTBD samples.

To generate microtubules for gliding assays, 2 µl of porcine tubulin (5 mg/ml) were mixed with 0.25 µl of TMR labelled tubulin (5 mg/ml, in general tubulin buffer, vendor: Cytoskeleton) and incubated for 5 min on ice. After the addition of 2.5 µl general tubulin buffer, 2.5 µl tubulin glycerol buffer and 0.75 µl 20 mM GTP (Cytoskeleton), the mixture was incubated at 37 °C for 20 min. To create the microtubule stock solution for the gliding assays, 25 µl of general tubulin buffer enriched with 20 µM taxol (cytoskeleton) were added.

The flow chamber was incubated with 10 µl anti-GFP (Roche, 500 µg/ml in PBS buffer) for 2 min and subsequently washed twice with 10 µl BRB80-Casein (80 mM NaOH PIPES pH 6.8, 2 mM $MgCl_2$, 1 mM EGTA, 1 mM DTT and 2 mg/ml Casein). $\text{Rea1}_{\Delta AAA2H2\alpha+\Delta4168-4907}$, $\text{Rea1}_{\Delta AAA2H2\alpha+\Delta4168-4907}$- spycatcher-dynein MTBD or a GFP labelled human dynein 2 motor domain construct[43] after gel filtration (0.15 mg/ml) was diluted 1:10 in BRB80 and 3 × 10 µl were applied to the flow chamber for 2 min followed by two 10 µl BRB80-Casein washing steps. The motility solution was prepared by adding 1 µl microtubule stock solution and 1 µl gloxy solution (catalase and glucose oxidase in BRB80 buffer[43]) to 98 µl BRB80-Casein, 3 mM Mg-ATP (or AMPPNP or without nucleotide), 40 mM KAc, 20 µM taxol, 0.5% glucose. 10 µl of the motility solution was applied to the flow chamber and after a 2 min incubation step the cover slide was imaged by TIRF microscopy. We collected a positive control with the human dynein2 construct to demonstrate the functionality of the microtubule gliding assay (Supplementary Movie 7). All movies were analysed with ImageJ (version 1.53n)[44].

Each of the four analysed conditions ($\text{Rea1}_{\Delta AAA2H2\alpha+\Delta4168-4907}$ -dynein MTDB + ATP; $\text{Rea1}_{\Delta AAA2H2\alpha+\Delta4168-4907}$ +dynein MTDB APO; $\text{Rea1}_{\Delta AAA2H2\alpha+\Delta4168-4907}$ +dynein MTDB + AMPPNP; $\text{Rea1}_{\Delta AAA2H2\alpha+\Delta4168-4907}$ +dynein MTDB + ATP) was imaged twice. For the statistical analysis, the field of view was separated into four equal areas, which were analysed separately leading to a total sample number of $n = 8$ ($2 \times 4$) per imaged condition. The typical observation time was between 30 and 60 min with a frame rate of 15 sec. The following definitions were used to characterize microtubules during the observation time: Diffusive: the microtubule detached from the cover slide; static: the microtubule remained attached to the cover slide; partially attached: only parts of the microtubule were attached to the cover slide; wiggling: the microtubule undergoes repeated wiggling along its long axis indicating force exposure; moving: the microtubule shows directed movement. To discriminate against microtubules that land on and subsequently simply slide along the cover slide till they reach a stable position, the event was only counted, if the microtubule remained static for 2 min before the start of the movement.

For the $\text{Rea1}_{\Delta AAA2H2\alpha+\Delta4168-4907}$ -dynein MTDB + ATP condition we observed: diffusive: 45.1% ± 11.2%; partially attached: 42.5% ± 14.2%; static: 12.0% ± 7.0%; wiggling: 0.4% ± 1.0%; moving: n.d. For the $\text{Rea1}_{\Delta AAA2H2\alpha+\Delta4168-4907}$ +dynein MTDB APO condition we observed: diffusive: n.d.; partially attached: 7.0 % ± 2.6%; static: 92.1% ± 2.6%; wiggling: 0.9 % ± 1.2%; moving: n.d. For the $\text{Rea1}_{\Delta AAA2H2\alpha+\Delta4168-4907}$ +dynein MTDB + AMPPNP condition we observed: diffusive: 7.3% ± 6.4%; partially attached: 32.0 % ± 8.2%; static: 59.9% ± 8.9%; wiggling: 0.8 % ± 0.7%; moving: n.d. For the $\text{Rea1}_{\Delta AAA2H2\alpha+\Delta4168-4907}$ + dynein MTDB + ATP condition we observed: diffusive: 5.8% ± 2.0%; partially attached: 40.8% ± 6.5%; static: 28.1% ± 4.5%; wiggling: 19.8% ± 4.4%; moving: 5.6% ± 1.3%. Microtubule gliding events were only observed in the presence of ATP and the dynein microtubule binding domain. Such events were not detected with motility solutions without Mg-ATP or with AMPPNP and samples lacking the dynein microtubule binding domain.

In order to demonstrate that the observed microtubule gliding events are caused by directed movement and not random diffusion, we analysed the mean squared displacement $<\Delta x^2>$ over time. In general, time dependent mean squared displacement is expected to follow the equation $<\Delta x^2> = D_\alpha * t^\alpha$, where $D_\alpha$ is the apparent diffusion coefficient and $\alpha$ is the anomalous coefficient. Random diffusion events are characterized by $\alpha = 1$, whereas directed movements are characterized by $\alpha > 1$. In order to determine $\alpha$, we used the logarithmic version of the equation above, $\log <\Delta x^2> = \log D_\alpha + \alpha * \log t$, and plotted $\log <\Delta x^2>$ vs $\log t$ for 19 individual microtubule gliding events. The slope of the fitted linear equation corresponds to $\alpha$. Averaging $\alpha$ for the $n = 19$ observed events leads to $\alpha = 1.49 \pm 0.22$. We carried out the equivalent analysis for $n = 11$ human dynein 2 microtubule gliding events. In this case we obtained $\alpha = 1.65 \pm 0.18$.

## ATPase assays
For all ATPase assays, the EnzChek Phosphate Assay Kit (Molecular Probes) was used according to the supplier recommendations. The reaction volume was 150 µl consisting of 30 µl 5x assay buffer (150 mM HEPES-NaOH pH 7.2, 10 mM Mg-Acetate, 5 mM EGTA, 50% Glycerol, 5 mM DTT), 30 µl MESG (EnzCheck substrate), 1.5 ml PNP (purine nucleoside phosphorylase), 10 µl Mg-ATP (15 mM) and 30 – 50 nM protein. Experiments were carried out on a GENios spectrophotometer (TECAN). The ATPase rates were 0.96 ± 0.17 and 0.54 ± 0.1 mol Phosphate/mol Rea1/s for $\text{Rea1}_{wt}$ and $\text{Rea1}_{D2915A-R2976A-D3042A}$ and determined in triplicate (Fig. 6e). For the nucleotide dependency of the ATPase activity of $\text{Rea1}_{wt}$ and $\text{Rea1}_{\Delta AAA2H2\alpha}$ we obtained: $\text{Rea1}_{wt}$ + ATP: 1.07 ± 0.03, $\text{Rea1}_{wt}$ + ATPγS: 0.07 ± 0.01, $\text{Rea1}_{wt}$ + AMPPNP: n.d., $\text{Rea1}_{\Delta AAA2H2\alpha}$ + ATP: 11.90 ± 0.73, $\text{Rea1}_{\Delta AAA2H2\alpha}$ + ATPγS: 0.42 ± 0.01, $\text{Rea1}_{\Delta AAA2H2\alpha}$ + AMPPNP: n.d. The $\text{Rea1}_{wt}$ + ATP assays were done in duplicate, the $\text{Rea1}_{wt}$ + ATPγS assays were done in quadruplicate and the $\text{Rea1}_{\Delta AAA2H2\alpha}$ + ATP as well as $\text{Rea1}_{wt}$ + ATPγS assays were determined five times (Supplementary Fig. 3).

## Determination of Rea1 mutant expression levels
Expression levels of $\text{Rea1}_{wt}$ and the $\text{Rea1}_{D2915A-R2976A-D3042A}$ mutant were assessed as follows. Cells expressing $\text{Rea1}_{wt}$ or $\text{Rea1}_{D2915A-R2976A-D3042A}$ were grown in 10 ml of YP medium (1% yeast extract, 1% peptone, Becton-Dickinson) supplemented with 2% glucose to OD600 = 0.2 and collected. Cell pellets were resuspended in 100 µl ice-cold MQ $H_2O$, transferred into ice-cold microtubes containing 100 µl Zirconia beads and 20 µl of 100% trichloroacetic acid (TCA). Cells were broken by vigorous agitation (vortex) at 4 °C, two times during 2 min separated by 1 min incubation on ice. 1 ml of ice-cold 5% TCA was added and samples were mixed with vortex for 30 sec. Samples were centrifuged 15 min at 11000 g and 4 °C in a benchtop centrifuge. Supernatants were removed and 80 µl of 2 x LDS NuPAGE loading buffer (Invitrogen, cat. # NP0007) supplemented with 100 mM dithiothreitol (DTT) were added to the cell/bead pellets followed by 25 µl of 1 M Tris-HCl, pH 9.5. Cell/bead pellets were resuspended by vigorous agitation 30 sec at 4 °C, spun down briefly and heated at 80 °C for 5 min. Samples were mixed by vigorous agitation 5 min at 4 °C, spun down briefly and heated again at 80 °C for 5 min. Samples were centrifuged 5 min at 11000 g and 4 °C in a benchtop centrifuge. Supernatants were collected and transferred to new tubes. Protein extracts were separated by SDS-PAGE using NuPAGE 3-8% Tris-Acetate gels (Invitrogen, cat. # EA0375BOX) in 1 x Tris-Acetate SDS running buffer (Invitrogen, cat. # LA0041) and the gels were stained using the SilverQuest staining kit (Invitrogen, cat. # LC6070)

## Miscellaneous
RNA extractions and northern blotting experiments were performed as described[45] using oligonucleotides 18S probe, 20S.3 probe, 23S.1 probe, rRNA2.1 probe and 25S probe (Supplementary Table 3). The anti-HA immunoprecipitation experiments were carried out as described in ref. 46. The western blot experiments in Supplementary

Fig. 14B were carried out as described in ref. 45. The anti-HA antibody (Roche Diagnostics, Cat. # 12013819001) was used at a 1:1000 dilution and the anti-Myc antibody (Covalab, ID Covalab: mab20008; Clone 9E10, batch number: 527700, Cat. # 00115009) was used at a concentration of 5 μg/ml.

## Reporting summary

Further information on research design is available in the Nature Portfolio Reporting Summary linked to this article.

## Data availability

The XL-MS dataset has been deposited on the ProteomeXchange Consortium via the PRIDE[41] repository with the dataset identifier PXD053636 and 10.6019/PXD053636 [https://proteomecentral. proteomexchange.org/cgi/GetDataset?ID=PXD053636]. The Mass Photometry raw and treated files generated in this study will be made fully available upon request. CryoEM maps generated in this study have been deposited at the Electron Microscopy Data Bank (EMDB) under accession codes EMD-50815, EMD-50816, EMD-50817 and EMD-50818. Source data are provided with this paper.

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

## Acknowledgements

This work was supported by the collaborative ANR PRC grant "Rea1Com" to H.S. (ANR-19-CE11-0011-01) and A.H., an ATIP-avenir grant to H.S. (CDP 0B1INSB-HS-9ADO1051) and an ANR grant to B.A. (ANR-21-CE12-0008-01). This study was further supported by the grant ANR-10-LABX-0030-INRT, a French State fund managed by the Agence Nationale de la Recherche under the frame program Investissements d'Avenir ANR-10-IDEX-0002–02. The authors acknowledge the support and the use of resources of the French Infrastructure for Integrated Structural Biology (FRISBI) ANR-10-INBS-05 and of Instruct-ERIC. This work was also supported by the CNRS, the University of Strasbourg, the "Agence National de la Recherche" and the French Proteomics Infrastructure (ProFI; ANR-10-INBS-08-03). H.G.F. acknowledges the French Ministry for Education and Research for funding of his PhD. This work of the Interdisciplinary Thematic Institute IMCBio, as part of the ITI 2021-2028 program of the University of Strasbourg, CNRS and Inserm, was supported by IdEx Unistra (ANR-10-IDEX-0002), and by SFRI-STRAT'US project (ANR 20-SFRI-0012) and EUR IMCBio (ANR-17-EURE-0023) under the framework of the French Investments for the Future Program. We thank Andrew Carter for providing a plasmid harboring the SRS-dynein MTBD construct. We also thank Nacho Molina for his advice on the rmsd vs time analysis of microtubule gliding events. We thank the IGBMC light microscopy (Elvire Guiot, Erwan Grandgirard and Bertrand Vernay) and electron microscopy (Alexandre Durand) platforms. We are grateful to Christine Maheu (CBI, Toulouse) and Jessie Bourdeaux (engineer in the Henry/Henras team) for expert technical assistance.

## Author contributions

J.B. collected, processed, analyzed, and interpreted negative stain and cryo-EM data; G.K. cloned Rea1 mutants, produced yeast strains, produced and purified Rea1 constructs, and carried out pre60S export assays; C.D. and B.A. cloned Rea1 mutants, produced yeast strains, performed northern blot experiments; T.K. and D.V. cloned and produced proteins for GST-pulldown experiments, and performed GST-pulldown experiments; P.S. produced yeast strains and established the pre60S export assay; N.M. provided reagents for cryoEM; C.C. collected negative stain EM data; H.G.F. designed, performed and analyzed data of crosslinking mass spectrometry and mass photometry experiments on Rea1 mutants; S.C. designed and supervised cross-linking mass spectrometry and mass photometry experiments on Rea1 mutants; Y.H. designed experiments, analysed data, produced yeast strains, carried out immunoprecipitation and northern blot experiments and analysed expression levels of Rea1 mutants; A.H. designed experiments, analysed data, produced yeast strains, performed northern blot experiments, carried out yeast-two-hybrid and co-immunoprecipitation experiments, and analysed expression levels of Rea1 mutants; H.S. produced yeast strains, designed experiments, analysed data and supervised the project; H.S., Y.H. and A.H. wrote the manuscript with input from all authors.

## Competing interests

The authors declare no competing interests.
