## [Peer Review File · Nature Communications]

Remodelling of Rea1 linker domain drives the removal of assembly factors from pre-ribosomal particlesReviewers' comments:

Reviewer #1 (Remarks to the Author):

Nearly a third of the 5000-residue six AAA+ domain containing 60S assembly factor Rea1 is made up of a multi-domain (stem, middle and top) linker. Further, a flexible and unstructured D/E rich region connects this linker to a MIDAS domain, which sequentially engages with assembly factors Rsa4 and Ytm1 and catalyses their removal 60S pre-ribosomes to promote progression toward export competence. How Rea1 extracts/pulls on the assembly factor is enigmatic, and also challenging to study.

In this manuscript, Busselez et al. focus on the importance of the linker domain in driving Rsa4 and Ytm1 release. The Rea1 linker was suggested to play a passive role to help the MIDAS domain capture Rsa4 and Ytm1 on the 60S pre-ribosome and drive their release in an ATP-dependent manner. They employ yeast genetics, biochemical purifications, and electron microscopy to develop a model of how the linker domain of Rea1 contributes to force generation in order to remove the assembly factors. Busselez et al. now provide evidence for a direct functional role of Rea1 linker region in assembly factor removal through a series of remodelling events, conformational changes, and gain/loss of interactions.

Overall, the study has been carefully crafted, well interpreted, and provides a new insight into the workings of Rea1 during 60S assembly. I have a few concerns listed below which the authors should address:

1. Can the authors quantify the abundance of the different conformational states that they observe and present the information directly in Fig 1E, Fig. 2, Fig. 4 and Fig. 6B.

2. The authors “observed occasional events of slow directed microtubule gliding over several minutes”. They interpret this as a remodelling of the linker top with respect to the AAA+ ring is able to produce mechanical force. Could the authors better describe this assay used and elaborate on the interpretation. I find this part of the manuscript weak. Given its importance to the main conclusion of the work, can the authors provide an alternative experiment to strengthen the point. For e. g. it may help to provide a positive control for the experiment where is faster microtubule gliding observed.

3. All the electron microscopy work has been performed on Rea1 and its mutant in absence of the 60S pre-ribosome. The different conformational states and their importance has been correlated to

how Rea1 might actually work on the 60S pre-ribosome. The Reviewer clearly understand the challenge to study Rea1, but it would still help (and not harm) to state the shortcomings of the work and where the interpretations might require further work.

4. The manuscript is very detailed and intense to read. There are clearly interesting observations which will attract interest from other fields, hence it would be helpful to focus on a few main points and find a way to bring it across to the general reader.

Reviewer #2 (Remarks to the Author):

In this work, Busselez and colleagues address the important question of how the ribosome maturation factor Rea1 fulfils its crucial precursor remodelling function during ribosome biogenesis. The authors suggest that a multi-step re-arrangement of the large Rea1 linker is essential for its function, whereas the ATP hydrolysis activity of the AAA+ ring plays only a minor role. They support their hypothesis structurally using negative stain and cryo EM analyses, biochemically with XLMS analyses, microtubule gliding assays, mutational analyses, and Northern analyses of pre-RNA, and in vivo with pre 60S ribosomal export assays. Despite not being able to obtain high-resolution cryo-EM structures of Rea1 they build a convincing case for a linker involvement in removal of assembly factors from pre60S particles. Overall the manuscript contains many original findings. The data are well presented and are comprehensive. The distribution of data between main text and supplementary material is appropriate. The most striking result is the effect of the linker mutants Rea1D2915A-R2976A-D3042A, Rea1 Δ 2916-2974, and Rea1 Δ 3072-3244, which exhibit complete or severe Rea1 dysfunction, arguing the case that the linker plays an essential role in Rea1 function. However, the authors should add a crucial control to their data and need to clarify some aspects of their hypothesis, which is not always presented in a comprehensive manner. Overall, this work will be of immediate interest to many people in the ribosome and AAA+ field and it represents an important scientific achievement justifying publication in Nature communications. However, a revision of the present manuscript and implementation of the control is strongly recommended.

Major points:

- The authors cannot rule out that some nucleotides are co-purified in their Rea1 preparations. They should sample the Rea1 2D classes under “absolute apo“ conditions after purification in a buffer without Mg and in the presence of at least 5mM EDTA to avoid co-purification of nucleotide.
- Linker remodelling is claimed to be a force producing event, but in which way the remodelling is dependent on ATP hydrolysis in the AAA+ ring remains unclear. Where does the force for linker and

substrate remodelling come from if not from the AAA+ activity? Furthermore, it is confusing to describe linker remodelling itself as a force producing event, since such a large Rea1 movement (which clearly only turns into a directed movement by ATP hydrolysis) is bound to show an effect in microtubule gliding assays, whether or not this movement is transmitted onto the substrate or the midas domain cannot be deferred from the experiments. In the end, it's the ATPase activity that produces the force.

- Hypothesis: Unfortunately, it is difficult to follow the authors' reasoning regarding the involvement of ATP hydrolysis and the role of AAA+ domains in Rea1 function. They seem to suggest that AAA+ force generation is not essential for assembly factor removal (page 13, line 12+13), but at the same time ATP hydrolysis and Rea1 function are impaired in their linker mutants and the last steps of linker remodelling are ATP dependent. The authors also state that the goal of linker remodeling is to bring domains at the top of the linker into contact with the AAA+ ring of the ribosome (page 12). However, the 2D classes of the nucleotide independent steps do not show a contact between the top of the linker and the AAA+ ring making a sensing function unlikely.

- The microtubule gliding assay indicates that ATP is required for linker remodeling because microtubule gliding occurs only in the presence of ATP. Contrary to the description in the manuscript, that Rea1 function is either dependent on linker remodelling or AAA+ ring driven remodelling, AAA+ activity and linker remodeling appear to be interrelated and both required for Rea1 function. This should be expressed more clearly in the manuscript.

- All FSC curves are cut at about 6A. It's impossible to judge whether the curves are actually falling to 0 and staying there at high resolution. Hence, the resolution estimation might be overestimated. The FSC curves should be extended to Nyquist frequency.

Minor points:

- The title of the manuscript is very confusing if not misleading. How can the linker remodeling be nucleotide independent and ATP driven?

- It would be very helpful if the manuscript would refer to the ATPase activity of the AAA+ domains; do all domains hydrolyse ATP? Is there any difference in the ATPase activity of the individual domains? Perhaps that can be incorporated into figure 1A?

(Spelling) errors:

- Page 2, line 9 insert "is" so that it reads: "...and is involved in the processing..."

- Page 4, line 29 should read conformations

- Page 15, line 22 delete "to the"

- Page 17, line 18 "*S. cerevisiae*" should be in italic

- Figure legend 5; 4th row from the bottom: delete "we"

Reviewer #3 (Remarks to the Author):

NCOMMS-23-08210-T

This manuscript by Busselez et al combines negative stain EM with some medium to low resolution cryo-EM and a few functional assays to study the mechanism of the Rea ATPase. None of the conclusions in this manuscript are supported by data. There are fundamental flaws with this work that preclude its publication. Even in a completely revised version its significance for Nature Communications is unclear.

Major points:

1. The analysis is based on the ATP-engaged state in 0.2-0.8% of the images and the authors analyze a few hundred images. That means they get a few images for each of these states. The absence of an image in that set-up means absolutely nothing. The authors need to analyze at least 5-10-fold more particles to draw any statistically significant conclusions.
2. Even if they can convince me that they can statistically reliably measure differences in these populations by obtaining 10x more data, why do we consider populations that are 0.2-0.8% of the population? I am certain there are many other things at that abundance. Other than that we like this model, there must be additional reasoning. This is exacerbated as the data actually make no sense. Why do they see these with ATP-gammaS, but not AMPPNP. Both are slowly-hydrolyzing ATP analogs. ATP-gammaS can also be more like ATP when you add Mn, or Zn, a soft metal ion. How do we know that these tiny populations do not arise from impurities in the nucleotide preps. Nucleotides are full of dirt from the synthesis and should be HPLC-purified (actually ATP has tons of ADP, and so has AMPPNP, and probably ADP-gammaS).
3. The conclusion that the engagement with the ring requires ATP hydrolysis is not just unsupported because of the issues with low sampling as above, but also the fact that ATP-gammaS, which is slowly hydrolyzing (and often considered a non-hydrolyzing analog) also has it. Moreover, if, as the authors contend at the end, engagement with the ring drives ATP hydrolysis, then this cannot be true (only one of these statements). AMPPNP is well-known to not always replicate the ATP-bound state. Sometimes AMPPCP does.
4. The functionality of the constructs in here should be tested. Not just the last two.
5. The Cryo-EM structure shows the extended conformation. The negative stain images do not recapitulate that and all show a "rotated" structure. First of all, this is extremely difficult to follow in the images. The two structures need to be presented side-by-side, with locations of rotation indicated and pivot points. Given that the "rotated" remodeled structure is based on 400 negative

stained images and probably at about 20 Å resolution, a robust mutational analysis must be carried out to support that structure. Also, the functionality of this structure must be demonstrated not just using the salt bridge mutant but such mutants that simply impair the “new” interface. Finally, how do we not know that one (or more) of the structures are artifacts from stain, interaction with the surface, air-water interface, etc.

6. The relationship between this “remodeling” in state 1 and the AAA-engagement or nucleotide states is not supported. There is no evidence that anything in the manuscript is nucleotide-dependent or informs about nucleotide-dependent remodeling. Thus, this seems to me like a collection of snapshots of a mobile molecule, where we have no idea if they represent any functional states.

7. The section on force-generation is completely overblown and lacks any controls. If they want to say anything about force, they need to show that this occurs with ATP, but not AMPPNP, or ADP. The difference between AMPPNP and ATP_γS must be also demonstrated in ATPase, remodeling and force assays.

8. The IP does not show anything. Not even the expression of the constructs (input) is the same, the “correct” thing is only observed as a very minor species in lane 2. And there is no reason to assume that the minor bands in the pulldown are the TOP2/3 and not other crossreacting species. This needs to be either totally redone, or removed.

9. Interactions the authors suggest exist need to be tested with mutagenesis, as the resolution of all the structures, even the best ones is at best moderate.

10. The only thing in the paper that might be a result are the findings with the salt bridge mutant. But, breaking something is very easy....can this be rescued if the salt bridges are reversed ie. Positive to negative and vice versa....importantly, I am also not convinced that the “remodeling” is actually critical, as it is observed without any nucleotide. That part needs to be better established. It just seems like wishful thinking.

11. The significance of the crosslinking data is unclear. How many total crosslinks are there? Where are all the other crosslinks? Are they all consistent with the structures? They should be mapped in ways that highlight their abundance. Crosslinking is notoriously non-specific.

12. The cryo-EM is lacking any documentation. How did the particles segregate? What fraction of the dataset was used?

To summarize, as far as I can tell there are negative stain images, but too much made of percentages that are unlikely to be statistically significant and not bolstered by enough data. Negative stain is also really low resolution and not suitable for nature communications. There are a few moderate resolution structures whose significance is unclear, which suggest a remodeling of the linker in a nucleotide-independent manner. The specific remodeling needs to be confirmed using mutagenesis. Such mutants could support their significance in addition to the specific suggested structures. The entire matter of the nucleotide-dependent engagement with the AAA-ring is not supported by the data. Overall, I cannot see how this work is a significant advance.

Reviewer #4 (Remarks to the Author):

Busselez et al. study the ribosome maturation factor Rea1 which catalyzes the removal of assembly factors from large ribosomal subunit precursors to promote their export from the nucleus to the cytosol. The authors demonstrate that Rea1 rotates and swings towards the AAA+ ring following a complex remodeling scheme involving nucleotide independent as well as nucleotide dependent steps. They thus conclude, (i) that ATP-hydrolysis is required to engage the linker with the AAA+ ring and ultimately with the AAA+ ring docked MIDAS domain and (ii) that the interaction between the linker top and the MIDAS domain allows force transmission for assembly factor removal.

To probe if the described linker remodeling is able to produce mechanical force, the authors carried out microtubule gliding assays. They modified Rea1 Δ AAA2H2 α + Δ 4168-4907 by fusing a GFP (allowing anchorage to the cover slip surface) to AAA5 and a spytag (with dynein microtubule binding domain) into the linker top2 domain. After the application of fluorescently labelled microtubules, they screened for microtubule gliding events. The authors observed occasional events of slow, directed microtubule gliding over several minutes (Figure 5C, Movies S3-S5) suggesting that the remodeling of the linker top with respect to the AAA+ ring is able to produce mechanical force. They reason that the mechanical force produced by linker remodeling might be directly applied to the AAA+ ring-docked MIDAS domain via the linker top2/top3 domains to remove assembly factors from pre60S particles.

While I am not an expert to judge the quality of the overall results of the manuscript (which I nevertheless find to be carried out on a very high level), I was asked by the editor to provide a brief technical report, assessing the reported TIRF microscopy based microtubule gliding assay experiments.

In general, I find it a great idea to design and perform these experiments. Also, I tend to believe that what the authors see is directional motility. However, to solidify their statement, the authors should convince the readers that the reported events are indeed directional translocation and not the result of random diffusive motion. After all, currently only three rare events are shown, out of fields of view where most of the microtubules appear to be statically attached to the surface or are genuinely diffusing. The authors could, for example, perform a mean-square displacement (MSD) analysis proving that MSD vs. time cannot be fit by a linear function alone - but rather needs to contain a quadratic function.

Moreover, it should be added to the manuscript: (i) the number of total gliding events, along with how many statically attached and purely diffusing microtubules have been observed, (ii) the gliding

velocities, (iii) scale bars in Figure 5C, and (iv) some discussion on the magnitude of the "forces". The forces needed to move microtubules as observed (short filaments with extremely low velocity on top of some random motion) are expected to be extremely low. Would these forces be sufficient to fulfill the ascribed function of assembly-factor removal?

Reviewer #1 (Remarks to the Author):

Nearly a third of the 5000-residue six AAA+ domain containing 60S assembly factor Rea1 is made up of a multi-domain (stem, middle and top) linker. Further, a flexible and unstructured D/E rich region connects this linker to a MIDAS domain, which sequentially engages with assembly factors Rsa4 and Ytm1 and catalyses their removal 60S pre-ribosomes to promote progression toward export competence. How Rea1 extracts/pulls on the assembly factor is enigmatic, and also challenging to study.

In this manuscript, Busselez et al. focus on the importance of the linker domain in driving Rsa4 and Ytm1 release. The Rea1 linker was suggested to play a passive role to help the MIDAS domain capture Rsa4 and Ytm1 on the 60S pre-ribosome and drive their release in an ATP-dependent manner. They employ yeast genetics, biochemical purifications, and electron microscopy to develop a model of how the linker domain of Rea1 contributes to force generation in order to remove the assembly factors. Busselez et al. now provide evidence for a direct functional role of Rea1 linker region in assembly factor removal through a series of remodelling events, conformational changes, and gain/loss of interactions.

Overall, the study has been carefully crafted, well interpreted, and provides a new insight into the workings of Rea1 during 60S assembly. I have a few concerns listed below which the authors should address:

1. Can the authors quantify the abundance of the different conformational states that they observe and present the information directly in Fig 1E, Fig. 2, Fig. 4 and Fig. 6B.

The revised figures 1E, 2 and 6B as well as the revised supplementary figures 2 and 4 show now the percentage of the total particles that sorted into the negative stain EM 2D class averages. We did not modify Figure 4, because it refers to the Rea1_{ΔAAH2α} ATPγS data set for which this information is already available in revised Figure 2.

2. The authors “observed occasional events of slow directed microtubule gliding over several minutes”. They interpret this as a remodelling of the linker top with respect to the AAA+ ring is able to produce mechanical force. Could the authors better describe this assay used and elaborate on the interpretation. I find this part of the manuscript weak. Given its importance to the main conclusion of the work, can the authors provide an alternative experiment to strengthen the point. For e. g. it may help to provide a positive control for the experiment where is faster microtubule gliding observed.

In the revised methods section, we now provide a more detailed description of the assays and how we analysed the data. We also have included additional control experiments and a statistical analysis to demonstrate that microtubule gliding events are only observed in the presence of ATP and the dynein microtubule binding domain (revised figure 5E). Furthermore, we provide an additional analysis demonstrating that the observed microtubule gliding events are due to directed movement and not random diffusion (revised figure 5F and G). At the request of the reviewer, we have also included movie S7, which shows a positive control with the microtubule motor protein dynein to demonstrate the functionality of the assays.

3. All the electron microscopy work has been performed on Rea1 and its mutant in absence of the 60S pre-ribosome. The different conformational states and their importance has been correlated to how Rea1 might actually work on the 60S pre-ribosome. The Reviewer clearly understand the challenge to study Rea1, but it would still help (and not harm) to state the shortcomings of the work and where the interpretations might require further work.

We have modified the revised manuscript accordingly. The revised discussion contains now a paragraph stating potential modifications/alterations of linker remodelling in the presence of pre60S particles (page 14, lines 3 - 11, revised manuscript).

4. The manuscript is very detailed and intense to read. There are clearly interesting observations which will attract interest from other fields, hence it would be helpful to focus on a few main points and find a way to bring it across to the general reader.

In order to reduce the amount of detail in the revised manuscript, we have removed the detailed description of the construct used for the microtubule gliding assays to the methods section. Furthermore, we have removed the negative stain EM and the functional data for the Rea1 Δ 2916-2974 and Rea1 Δ 3072-3244 mutants, respectively. In the revised manuscript, we focus now on the Rea1 Δ 2915A-R2976A-D3042A, which is the best characterized mutant and supports all the major conclusions of the manuscript.

Reviewer #2 (Remarks to the Author):

In this work, Busselez and colleagues address the important question of how the ribosome maturation factor Rea1 fulfils its crucial precursor remodelling function during ribosome biogenesis. The authors suggest that a multi-step re-arrangement of the large Rea1 linker is essential for its function, whereas the ATP hydrolysis activity of the AAA+ ring plays only a minor role. They support their hypothesis structurally using negative stain and cryo EM analyses, biochemically with XLMS analyses, microtubule gliding assays, mutational analyses, and Northern analyses of pre-RNA, and in vivo with pre 60S ribosomal export assays. Despite not being able to obtain high-resolution cryo-EM structures of Rea1 they build a convincing case for a linker involvement in removal of assembly factors from pre60S particles. Overall the manuscript contains many original findings. The data are well presented and are comprehensive. The distribution of data between main text and supplementary material is appropriate. The most striking result is the effect of the linker mutants Rea1 Δ 2915A-R2976A-D3042A, Rea1 Δ 2916-2974, and Rea1 Δ 3072-3244, which exhibit complete or severe Rea1 dysfunction, arguing the case that the linker plays an essential role in Rea1 function. However, the authors should add a crucial control to their data and need to clarify some aspects of their hypothesis, which is not always presented in a comprehensive manner. Overall, this work will be of immediate interest to many people in the ribosome and AAA+ field and it represents an important scientific achievement justifying publication in Nature communications. However, a revision of the present manuscript and implementation of the control is strongly recommended.

Major points:

- The authors cannot rule out that some nucleotides are co-purified in their Rea1 preparations.

They should sample the Rea1 2D classes under “absolute apo” conditions after purification in a buffer without Mg and in the presence of at least 5mM EDTA to avoid co-purification of nucleotide.

We have carried out the requested control experiment and included the results in the revised manuscript (page 5, lines 6 - 10, supplementary figure 2). We identify the same extended and intermediate linker remodelling states as reported for the original apo data set (Figure 1E, original manuscript) again indicating that the observed remodelling states 1 – 5 do not depend on nucleotide and are part of the intrinsic conformational flexibility of Rea1.

- Linker remodelling is claimed to be a force producing event, but in which way the remodelling is dependent on ATP hydrolysis in the AAA+ ring remains unclear. Where does the force for linker and substrate remodelling come from if not from the AAA+ activity? Furthermore, it is confusing to describe linker remodelling itself as a force producing event, since such a large Rea1 movement (which clearly only turns into a directed movement by ATP hydrolysis) is bound to show an effect in microtubule gliding assays, whether or not this movement is transmitted onto the substrate or the midas domain cannot be deferred from the experiments. In the end, it's the ATPase activity that produces the force.

We agree with the reviewer that this aspect was not well explained in the original manuscript. Of course, ultimately the force for assembly factor removal will be generated via ATP hydrolysis in the AAA+ ring.

One of the central questions we are addressing in this manuscript is whether ATP-hydrolysis driven conformational changes in the AAA+ ring are directly or indirectly transformed into force for assembly factor removal. Our results suggest that the latter is the case. ATP hydrolysis in the Rea1 AAA+ ring drives the remodelling of the linker (states 6, 7 and 8), which in turn produces the force for assembly factor removal.

In the revised manuscript, we now clearly state that only the ATP-hydrolysis driven steps of Rea1 linker remodelling - states 6, 7 and 8 – are the ones that are associated with force generation (page 9, lines 28 – 31, revised manuscript).

The reviewer is also right to point out that our microtubule gliding experiments do not prove that the force generated by linker remodelling is transmitted to the MIDAS domain. We have mentioned this aspect as caveat in our revised discussion (page 13, lines 29 – 31, revised manuscript). However, it was also not our intention to demonstrate force transmission to the MIDAS domain. We see the microtubule gliding assays as “proof-of-principle” experiments that for first time show that ATP-hydrolysis dependent linker remodelling with respect to the AAA+ ring is able to produce mechanical force in general.

- Hypothesis: Unfortunately, it is difficult to follow the authors' reasoning regarding the involvement of ATP hydrolysis and the role of AAA+ domains in Rea1 function. They seem to suggest that AAA+ force generation is not essential for assembly factor removal (page 13, line 12+13), but at the same time ATP hydrolysis and Rea1 function are impaired in their linker mutants and the last steps of linker remodelling are ATP dependent. The authors also state that the goal of linker remodeling is to bring domains at the top of the linker into contact with the

AAA+ ring of the ribosome (page 12). However, the 2D classes of the nucleotide independent steps do not show a contact between the top of the linker and the AAA+ ring making a sensing function unlikely.

As outlined in the previous point, we have now clarified the relationship between ATP-hydrolysis in the AAA+ ring, linker remodelling and force generation for assembly factor removal from pre60S particles in the revised manuscript. We also have modified a paragraph in the discussion of the revised manuscript to clearly state that ATP hydrolysis in the Rea1 AAA+ ring is always required to produce the force for assembly factor removal. The interesting mechanistic question is if the ATP hydrolysis driven conformational changes in the AAA+ ring directly lead to force production or if they are harnessed to remodel the linker, which in turn produces the force for assembly factor removal (page 15, lines 1 – 11, revised manuscript).

Concerning the “sensing function” of the nucleotide independent linker remodelling steps that we mentioned in the discussion section, we still feel that this is a possibility that is worthwhile mentioning. In our opinion a potential sensing function of the nucleotide independent linker remodelling steps does not require the linker to contact the AAA+ ring. It is sufficient if the linker is in close proximity to the AAA+ ring. Depending whether or not essential maturation events have occurred, the pre60S particle environment could stimulate the ATPase activity of the AAA+ ring in such a way that in turn allows the ATP-hydrolysis dependent AAA+ ring engagement of the linker for assembly factor removal via states 6, 7 and 8. We have modified the corresponding paragraph in the discussion (page 14, lines 23 – 29, revised manuscript).

- The microtubule gliding assay indicates that ATP is required for linker remodeling because microtubule gliding occurs only in the presence of ATP. Contrary to the description in the manuscript, that Rea1 function is either dependent on linker remodelling or AAA+ ring driven remodelling, AAA+ activity and linker remodeling appear to be interrelated and both required for Rea1 function. This should be expressed more clearly in the manuscript.

Throughout the revised manuscript we now clearly state that the ATP hydrolysis associated linker remodelling states 6,7 and 8 depend on ATP-hydrolysis in the Rea1 AAA+ ring.

- All FSC curves are cut at about 6Å. It's impossible to judge whether the curves are actually falling to 0 and staying there at high resolution. Hence, the resolution estimation might be overestimated. The FSC curves should be extended to Nyquist frequency.

The reviewer expresses here concerns about the real resolution of the cryoEM maps as indicated by the FSC curves. It is true that relion sometimes reports inflated nominal resolution estimates and we were aware of this issue already in our original manuscript. That is why we carefully did not report the nominal resolution limits in the main manuscript. Instead, we give the reader a more qualitative description of the obtained cryoEM maps (“medium and low resolution maps”, “secondary structure elements are resolved”, “individual AAA+ domains can be docked”, “the stem-AAA+ ring region can be docked as rigid body”). We also provide the revised supplementary figures 6 and 14, which demonstrate the fit between the obtained cryoEM maps and the corresponding structural models so that the reader can judge the quality of the cryoEM maps.

We would like to point out that none of the conclusions drawn from the presented cryoEM maps require high-resolution information. The Rea1 Δ AAA2H2 α ATPyS cryoEM map is actually low pass filtered to 25 Å before it is used to assign individual Rea1 subdomains in the negative stain EM 2D

class averages (revised figure 3C). Conformation II and III of the Rea1_{D2915A-R2976A-D3042A} mutant are used to support the statement that linker remodelling occurs at the interface between the linker middle and stem domains (revised figure 6F and revised supplementary figure 15). Even though these are low resolution maps, their quality is sufficient to dock the top-middle domain region of the linker as well as the linker-stem-AAA+ ring region as two individual rigid bodies (revised supplementary figure 14 B and C). Furthermore, it is highly unlikely that using the unbinned data will lead to new structural insights or improve the resolution of the maps as outlined below.

Following common practise in the field, we binned the cryoEM data to speed up image processing. Since the data was collected at a pixel size of 0.862 Å/pix and binned 4x, the pixel size of the binned data is 3.45 Å/pixel. That means that the theoretical resolution limit for our binned data sets (Nyquist frequency) is around 7 Å (2 x 3.45 Å). In case of the cryoEM maps of Rea1_{ΔAAA2H2α} ATPγS and Rea1_{D2915A-R2976A-D3042A} conformation I, the nominal resolution reported by relion of around 7 Å reflects the theoretical Nyquist frequency resolution limit. Also the real quality of the corresponding cryoEM maps corresponds to 7 Å data, because α-helices can clearly be resolved (compare revised supplementary figures 6B and 14A), which is expected for this resolution range. Using the unbinned data might improve the resolution further, even though it is highly unlikely that near atomic resolution can be reached. However, since these structures represent the already known straight linker conformation - for which already multiple high-resolution structures are available in the protein data bank – it is highly unlikely they will lead to new structural insights even if near atomic resolution could be achieved in the end. Again, high resolution is not needed to support any of the conclusions in the manuscript.

In the case of the alternative linker conformations II and III of the Rea1_{D2915A-R2976A-D3042A} mutant the reported nominal resolution by relion is also around 7 Å (compare revised supplementary figures 14 B and C). However, this seems to be indeed inflated as the corresponding maps do not show individual α-helices (compare revised supplementary figures 14 B and C). The real resolution of the maps is clearly much lower than 7 Å. Using the unbinned data in these cases will not improve the quality of the maps, because the binned data does even not reach the theoretical limit of 7 Å. Mostly likely it would only add noise.

For all these reasons, we would like to stay with the cryoEM maps and FCS curves calculated from the binned data.

Minor points:

- The title of the manuscript is very confusing if not misleading. How can the linker remodeling be nucleotide independent and ATP driven?

We have revised the title of the manuscript to: “Nucleotide independent as well as ATP-hydrolysis driven steps of Rea1 Linker remodelling drive the removal of assembly factors from pre-ribosomal particles”

- It would be very helpful if the manuscript would refer to the ATPase activity of the AAA+ domains; do all domains hydrolyse ATP? Is there any difference in the ATPase activity of the individual domains? Perhaps that can be incorporated into figure 1A?

We have included the requested information in the introduction (page 2, lines 34 + page 3, lines 1 – 2, revised manuscript)

(Spelling) errors:

- Page 2, line 9 insert “is” so that it reads: “...and is involved in the processing...”
- Page 4, line 29 should read conformations
- Page 15, line 22 delete “to the”
- Page 17, line 18 “*S. cerevisiae*” should be in italic
- Figure legend 5; 4th row from the bottom: delete “we”

This has been done

Reviewer #3 (Remarks to the Author):

NCOMMS-23-08210-T

This manuscript by Busselez et al combines negative stain EM with some medium to low resolution cryo-EM and a few functional assays to study the mechanism of the Rea ATPase. None of the conclusions in this manuscript are supported by data. There are **fundamental flaws** with this work that preclude its publication. Even in a completely revised version its significance for Nature Communications is unclear.

Major points:

1. The analysis is based on the ATP-engaged state in 0.2-0.8% of the images and the authors analyze a few hundred images. That means they get a few images for each of these states. The absence of an image in that set-up means absolutely nothing. The authors need to analyze at least 5-10-fold more particles to draw any statistically significant conclusions.

The reviewer essentially challenges that the size of the negative stain data sets is sufficient to support the hypothesis that AAA+ ring engagement of the linker (states 6, 7 and 8, Figure 2, original manuscript) is driven by ATP hydrolysis. The argument is that the low abundance, ATP-hydrolysis driven linker remodelling states 6, 7 and 8 would also be detected in the absence of nucleotide if larger data sets were collected.

Concerning the size of our negative stain EM data sets our rationale was to aim for at least 70K – 80K particles since these data set sizes allowed the reproducible detection of the rare “AAA+ ring engaged” linker remodelling states under ATP/ATP_γS hydrolysis conditions (Rea1_{ΔAAA2H2α} ATP_γS: 74141 particles; Rea1_{wt} ATP: 70952 particles; Rea1_{ΔAAA2H2α} ATP_γS + PhoX: 85123 particles, Rea1_{wt} ATP_γS: 87826 particles; original manuscript, page 18 lines 6 – 11, Figures 1E and 2, supplementary Figure 9B, original manuscript). We are surprised that the fact that we were able to reproducibly

detect the rare “AAA+ ring engaged” linker remodelling states in these four independently collected negative stain EM data sets was not acknowledged by the reviewer.

Furthermore, we reproducibly did not detect the AAA+ ring engaged states under non ATP-hydrolysis conditions in five independently collected data sets (Rea1_{wt} APO: 69908 particles; Rea1_{wt} AMPPNP: 76407 particles; Rea1_{ΔAAA2H2α} ADP: 97479 particles; Rea1_{ΔAAA2H2α} AMPPNP: 93919 particles; Rea1_{ΔAAA2H2α} APO: 314443 particles; original manuscript, Figures 1E and 2, supplementary Figure 2, page 18 lines 6 - 11). Combined these data sets represent 652156 particles, ≈ 9 times the size of the ATPγS data set. Again, this was not acknowledged by the reviewer.

Having outlined our rationale for the negative stain EM data set sizes above, we were wondering what the rationale of the reviewer was to demand 5 – 10 times more data? On which measure is this demand based? How has the reviewer concluded that simply doubling the data is not sufficient and collecting 20-times more data is not necessary? We believe that in a high-quality journal like “Nature Communications” such demands should be based on an objective measure rather than the “gut feeling” of the reviewer.

In fact, how arbitrary this demands seems to be is revealed by the reviewer him/herself. While – according to the suggestions of the reviewer – collecting 5 times more data would be sufficient for point 1, the demand is already raised to 10 times more data in point 2. The reviewer also does not specify, for which nucleotide states the additional data is demanded. We assume the reviewer does not suggest to collect more data on the ATP/ATPγS conditions since the rare states are already detected with the current data set sizes. Since the reviewer does not believe that the rare AAA+ ring engaged linker states dependent on the presence of nucleotide (point 3), we assume that a large apo data set must be a key priority for the reviewer.

It is then surprising that it apparently has escaped the attention of the reviewer that the data for such a control experiment is already included in the original manuscript. The AAA+ ring engaged linker states 6, 7 and 8 could be detected in our Rea1_{ΔAAA2H2α} ATPγS data set (Figure 2, original manuscript), but not in the Rea1_{ΔAAA2H2α} APO data set (Supplementary figure 2, original manuscript) even though this data set consisted of ≈ 4.5 times more particles (74141 for the Rea1_{ΔAAA2H2α} ATPγS data set vs 314443 for the Rea1_{ΔAAA2H2α} APO data set, page 18, lines 6 – 11, original manuscript). Why this was not acknowledged by the reviewer is puzzling to us.

To further strengthen the point that nucleotide binding is not sufficient to induce the “AAA+ engaged” linker states, we collected a larger data set of Rea1_{ΔAAA2H2α} in the presence of the non-hydrolyzable ATP analogue AMPPNP. At the request of the reviewer, this data set consisted of 387291 particles, ≈ 5 times more data compared to Rea1_{ΔAAA2H2α} ATPγS data set (page 21, lines 8 - 9, revised manuscript). Also in this case we did not detect the ATP hydrolysis driven, “AAA+ ring engaged” states 6, 7 or 8 (page 5, lines 31 - 33, revised manuscript).

2. Even if they can convince me that they can statistically reliably measure differences in these populations by obtaining 10x more data, why do we consider populations that are 0.2-0.8% of the population? I am certain there are many other things at that abundance. Other than that we like this model, there must be additional reasoning. This is exacerbated as the data actually make no sense. Why do they see these with ATP-γS, but not AMPPNP. Both are slowly-hydrolyzing ATP analogs. ATP-γS can also be more like ATP when you add Mn, or Zn, a soft metal ion. How do we know that these tiny populations do not arise from impurities in the nucleotide preps.

Nucleotides are full of dirt from the synthesis and should be HPLC-purified (actually ATP has tons of ADP, and so has AMPPNP, and probably ADP-gammaS).

The low abundance of certain conformations does not exclude the possibility that they are functionally important. The reviewer should provide more substantial reasons for why he/she questions the functional importance of these states. Setting a cut off percentage for the functional importance of Rea1 conformations (or any other protein conformation) is in our opinion very problematic if not impossible. What would be an acceptable number for the reviewer? 1%, 5% or 25%? On which rationale would such a threshold be based? Would it only apply to Rea1 or does the reviewer believe there is a general threshold for all proteins?

Concerning the “additional reasoning” for why we think that the low abundance, AAA+ ring engaged linker states are indeed functionally important, we refer to the key information in the manuscript:

-Functional studies by the Hurt lab established that the Rea1 mediated removal of Ytm1 and Rsa4 from pre60S particles requires ATP hydrolysis. Rea1 mediated Ytm1 and Rsa4 removal was not detected in the presence of the non-hydrolysable ATP analogue AMPPNP, suggesting that there are functionally important Rea1 conformations that are associated with ATP-hydrolysis (Introduction, original manuscript).

-Comparing negative stain data sets collected in the presence of ATP or the slowly hydrolysable ATP-analogue ATPγS to negative stain data sets collected under APO, ADP and AMPPNP conditions suggests that there at least three “AAA+ ring engaged” linker states associated with ATP-hydrolysis: 6, 7 and 8 (Figure 2 and supplementary Figure 2, original manuscript). Combined with the previous point this alone already suggests that they are functionally important.

-In the final linker remodelling state detected in our data, state 8, the linker top2/3 domains are in close proximity to the AAA+ ring docked MIDAS domain (Figure 4A and B, original manuscript), which suggests that there is an interaction between these Rea1 domains. We provide additional support for this interaction by complementary methods: immunoprecipitation (Supplementary figure 10B, original manuscript), crosslinking massspectrometry (Figure 5A, original manuscript) and in-vitro pulldown assays of recombinantly expressed and purified top2/3 and MIDAS domains (Figure 5B, revised manuscript). Importantly, the crosslinking mass spec data demonstrates that a highly conserved MIDAS domain loop region (amino-acids residues E4656-K4700) interacts with the linker top2 domains. This loop region is not involved in MIDAS domain mediated Rsa4 or Ytm1 binding (Ahmet et al., 2019, reference 20, original manuscript), but nevertheless was found to be essential for the removal of the assembly factor Rsa4 from pre60S particles in in-vitro release assays (Ahmet et al., 2019, reference 20, original manuscript). These findings suggest an indirect role of the E4656-K4700 loop region in assembly factor removal. The data presented in our manuscript suggest the molecular basis for this indirect role: The nucleotide dependent remodelling of the Rea1 linker allows the MIDAS domain engaged top2 domain to pull on the MIDAS domain and so indirectly on the MIDAS domain engaged Ytm1 and Rsa4 assembly factors to remove them from pre60S particles. This clearly provides strong additional support for the functional relevance of state 8 and the ATP hydrolysis driven linker remodelling states 6 and 7 that lead up to state 8. The data support a model where ATP hydrolysis drives the engagement of the linker top with the AAA+ ring and ultimately the AAA+ ring docked MIDAS domain (state 8) to pull on the MIDAS domain to remove Ytm1 or Rsa4 from pre60S particles.

-Furthermore, we present negative stain EM and functional data on Rea1 mutants (Figure 6B and Figure 7, original manuscript). The negative stain data of the Rea1 saltbridge mutant shows that ATP hydrolysis does not lead anymore to linker - AAA+ ring engagement and as a consequence the AAA+ ring engaged linker remodelling states 6,7 and 8 could not be detected. The functional analysis of the

Rea1 salt bridge mutant reveals that there are rRNA processing defects as a consequence of impaired Ytm1 removal and nuclear export defects of pre60S particles. These results provide additional support for the functional relevance of the AAA+ ring engaged linker remodelling states, because their absence in mutants correlates with severe functional defects.

It is new to us that both, ATP γ S and AMPPNP, should be hydrolysed by AAA+ proteins. The consensus in the AAA+ field seems to be to regard ATP γ S as slowly hydrolysing ATP analogue and AMPPNP as non-hydrolysable ATP analogue. In order to ultimately resolve this issue, we have carried out ATPase assays in the presence of ATP γ S and AMPPNP that clearly show that Rea1 hydrolyses ATP γ S at 4-6% of the ATPase rate, whereas in the case of AMPPNP hydrolysis is not detected (Supplementary figure 3, revised manuscript). We would like to clearly state here that these new data are in complete agreement with our observation in the negative stain data sets, e.g. in the presence of ATP γ S we do detect the hydrolysis driven AAA+ ring engaged linker states, whereas we do not detect these states in the presence of AMPPNP. The statement of the reviewer that “the data actually make no sense” is clearly wrong.

Concerning the purity of the nucleotides used in our study, we would like to point out that we always used commercially available, HPLC purified samples.

3. The conclusion that the engagement with the ring requires ATP hydrolysis is not just unsupported because of the issues with low sampling as above, but also the fact that ATP- γ S, which is slowly hydrolyzing (and often considered a non-hydrolyzing analog) also has it. Moreover, if, as the authors contend at the end, engagement with the ring drives ATP hydrolysis, then this cannot be true (only one of these statements). AMPPNP is well-known to not always replicate the ATP-bound state. Sometimes AMPPCP does.

We would like to point out that in our opinion the criticism is not well explained and hard to understand. The reviewer mentions “engagement with the ring”, but does not further specify this “engagement”. We believe she/he is referring to the linker engagement with the AAA+ ring. The argument of the reviewer seems to be that the AAA+ ring engagement of the Rea1 linker cannot be driven by ATP-hydrolysis, because we detect these states also in the presence of ATP γ S, which indirectly suggests that in his/her opinion ATP γ S cannot induce the conformational changes that would normally be driven by ATP-hydrolysis (although not explicitly stated by the reviewer). Why ATP γ S, which as we show in supplementary figure 3 (revised manuscript) can slowly be hydrolysed by Rea1, should not be able to drive conformational changes that would also occur with ATP-hydrolysis is not explained by the reviewer.

There are numerous examples in the AAA+ literature where ATP γ S has been used in functional/structural studies to investigate the mechanism of AAA+ proteins. On page 5 line 25 of the original manuscript, we cite an example where ATP γ S has been used to enrich transient, mechanistically relevant protein conformations (reference 24, original manuscript). There is no reason why such an approach should not work in the case of Rea1. Furthermore, we structurally investigated the ATP and ATP γ S states of Rea1 (Figure 2, original manuscript) and detect in both cases the ATP-hydrolysis driven, AAA+ ring engaged linker states 6 and 7. These results clearly support the idea that ATP γ S can induce conformational changes that also occur under ATP-hydrolysis.

As additional argument to challenge the statement that ATP-hydrolysis drives the AAA+ ring engagement of the linker, the reviewer refers to our interpretation of the mutant ATPase data in Figure 6E (original manuscript). Based on this data we suggest that linker engagement with the AAA+

ring stimulates the ATP-hydrolysis activity. Again, the rationale for this criticism remains vague. The reviewer does not explain why the idea of stimulated ATP-hydrolysis activity by the AAA+ ring engaged linker should exclude the possibility that ATP-hydrolysis in the AAA+ ring drives linker engagement. These are not mutually exclusive events. ATP-hydrolysis can induce AAA+ ring engagement of the linker and the AAA+ ring engaged linker in turn can stimulate the ATPase activity of the AAA+ ring, for example by inducing conformational changes in the AAA+ ring that accelerate the release of the hydrolysis products. We have clarified this aspect in the discussion section of our revised manuscript (page 13, lines 15 – 19, revised manuscript).

Concerning the question if AMPPNP mimics the ATP bound state in AAA+ proteins the reviewer is right to point out that this is not always the case. To resolve this issue for Rea1, we have generated a Walker-B mutant and carried out our negative stain EM analysis in the presence of ATP, which is the standard strategy for AAA+ proteins in such cases. We find no difference between the AMPPNP state and the ATP state of the Walker-B mutant, suggesting that AMPPNP mimics the ATP bound state in the case of Rea1. We have included the results of these additional experiments in the revised manuscript (Figure 2 and page 5 lines 33-34 + page 6 lines 1 - 4, revised manuscript).

4. The functionality of the constructs in here should be tested. Not just the last two.

In the revised manuscript we now focus on the Rea1_{D2915A-R2976A-D3042A}, salt-bridge mutant (compare our response to point 4 of reviewer 1), for which all functional data is available (ATPase activity, tetrad dissection, rRNA processing analysis, assembly factor release and pre60S particle export assays).

5. The Cryo-EM structure shows the extended conformation. The negative stain images do not recapitulate that and all show a “rotated” structure. First of all, this is extremely difficult to follow in the images. The two structures need to be presented side-by-side, with locations of rotation indicated and pivot points. Given that the “rotated” remodeled structure is based on 400 negative stained images and probably at about 20 Å resolution, a robust mutational analysis must be carried out to support that structure. Also, the functionality of this structure must be demonstrated not just using the salt bridge mutant but such mutants that simply impair the “new” interface. Finally, how do we not know that one (or more) of the structures are artifacts from stain, interaction with the surface, air-water interface, etc.

We agree with the reviewer that the relation between cryoEM and the negative stain data could be better explained in the manuscript. To this end we have now also included a 3D reconstruction of the ATPyS negative stain data set and show a comparison with the corresponding cryoEM structure (Supplementary figure 8, revised manuscript). The comparison shows a high similarity between the structures, which are both clearly in the extended, straight linker state. In “3D” cryo and negative data are consistent, which is also in agreement with our earlier work on Rea1 (Sosnowski et al., 2018).

The fact that we provide nevertheless strong evidence for the alternative linker states 1 – 8 by the negative stain 2D class averages in figure 2 of our original manuscript does not contradict these results. Please note the difference between “3D” and “2D” in electron microscopy. We refer the reviewer to page 4 lines 18 -24 of our original manuscript where we lay out the reasons for why we decided to investigate Rea1 linker remodelling by negative stain electron microscopy. With respect to the “3D-2D issue” we state: “We also limited our image processing workflow at the 2D classification

stage to avoid failing to detect alternative linker conformation during classification due to insufficient 2D projection distributions”, which is exactly the reason for the difference between “3D” and “2D” in case of the negative stain data set. In order to obtain a convincing 3D reconstruction, particles from 2D class averages representing many different 2D projections of the object under study have to be combined. The 2D class averages we show in figure 2 of the original manuscript are just a subset of the total 2D class averages that we detected in the negative stain data set (compare supplementary figure 16B, original manuscript). The majority of these 2D class averages represent different 2D projections of the Rea1 conformation in the straight linker state, which is why we were able to obtain a convincing 3D reconstruction for the extended linker state. For the Rea1 conformations in the alternative linker states 1 – 8 represented by the 2D class averages shown in figure 2, we do not have sufficient alternative 2D projections to obtain convincing 3D reconstructions. In fact, we strongly believe that each of these alternative linker states is just represented by the single 2D class average we show in figure 2, which means that it is impossible to obtain the corresponding 3D reconstruction.

The reviewer also asks for a side-by-side comparison of structures, without specifying which structures she/he has in mind. It would have been helpful if a reference to the manuscript (figure or page and line number) had been included in the reviewer comment. We believe the reviewer suggests to compare the cryoEM structure to one of the alternative linker states represented by the negative stain 2D class averages in figure 2 of the original manuscript. Since it is impossible to do a side-by-side comparison of a 3D cryoEM structure with a 2D class average, we assume the reviewer refers to our composite 3D model for the alternative linker remodelling state 1 that we present in supplementary figure 6A (original manuscript).

We are surprised that the reviewer is asking for locations of rotations and pivot points, because this information is already included in supplementary figure 6B (original manuscript), which shows the comparison of the cryoEM structure and the composite model. The reviewer is wrong to assume that the resolution of the composite model is 20 Å. Since we used the cryoEM map to create the composite model, it has the same resolution as the cryoEM structure, which is around 7 Å (supplementary figure 4, original manuscript). Concerning the validation of the composite model, the reviewer has not acknowledged that such validation had been already included in the original manuscript. In supplementary figure 6A (original manuscript) we demonstrate that the 2D projection of our composite model for linker remodelling state 1 is in agreement with the experimental negative EM 2D class average for this state. We feel that this validation is sufficient. The whole purpose of the composite model was to provide the reader with an estimate of the magnitude of conformational changes between the straight linker conformation and linker remodelling state 1, which is not essential for any of the conclusions in the manuscript. That is also why we included the composite model as supplementary item and not as main text figure.

We were pleased to read that the reviewer rightfully acknowledges that the functional relevance of linker remodelling state 1 is already demonstrated by the salt-bridge mutant data (Figures 6 and 7, original manuscript). Nevertheless, the reviewer demands additional functional investigations on mutants that essentially destabilize Rea1 linker remodelling state 1. In our opinion that will first require an experimentally determined cryoEM structure of this state with sufficient resolution to resolve side-chains. This would allow us to carefully analyse the interface between the linker stem and middle domains in order to identify promising sites for mutagenesis. From the composite model alone, there are no obvious candidates and a simple trial-and-error approach will have extremely low chances of success. Since linker remodelling state 1 is part of the intrinsic conformational flexibility of Rea1 and not dependent on a certain nucleotide state, it will be quite challenging to sufficiently stabilize this state for structural high-resolution investigations. We most likely will also encounter the

additional challenges of low abundance and/or preferred orientations. In light of these various challenges we feel that pushing for a high-resolution structure of state 1 would be clearly beyond the scope of this manuscript.

Concerning the question if the alternative linker remodelling states could result from negative staining artefacts, there are multiple lines of evidence that clearly show that this is not the case:

Firstly, the alternative linker remodelling states 1 – 8 are non-random, they are all related by the swing of the linker towards the AAA+ ring and its rotation around the long linker axis. It is highly unlikely – if not impossible – that a random negative staining artefact will produce a whole series of eight non-random conformations.

Secondly, we demonstrate that the 3D reconstructions of the straight linker conformation for the cryoEM and negative stain data sets are highly similar (compare supplementary figure 8, revised manuscript), which also clearly argues against artefacts introduced by negative staining.

Thirdly, 2D projections of a Rea1 cryoEM structure match up with Rea1 negative stain 2D class averages of the alternative linker states (Figures 3C, 4C and 4D, original manuscript), also clearly indicating that the latter are not compromised by negative staining.

Fourthly, we observe nucleotide dependent conformational changes. States 6, 7 and 8 are only observed under ATP/ATP γ S hydrolysis conditions and not in the absence of nucleotide or the presence of ADP or AMPPNP. If the linker remodelling states we observe would be staining artefacts, we should always detect the same states independent of the nucleotide condition, which is clearly not the case.

Fifthly, in addition to the nucleotide specific differences, we are also able to detect specific differences between constructs. The Rea1 wild type linker remodelling states do not harbour the AAA+ ring docked MIDAS domain (Figures 1E, 3D and supplementary figure 7, original manuscript). The linker states of the Rea1 Δ AAA2H2 α construct do have a AAA+ ring docked MIDAS domain (Figures 2, 3C and 3D, original manuscript). The Rea1_{D2915A-R2976A-D3042A} mutant linker states do not show the rotation described for Rea1 wild type and Rea1 Δ AAA2H2 α (Figure 6B, original manuscript). If the linker states would only be induced by the stain, we would not expect to see such differences for Rea1 mutants.

Sixthly, the negative stain EM analysis of crosslinked and non-crosslinked samples demonstrates identical sets of linker remodelling states (Supplementary figure 9B, original manuscript). If what we observed would be induced by the staining procedure it should make a difference if the protein is crosslinked or not before it is stained. The crosslinked sample would be resistant to the artefact induced by the stain which should lead to differences in the detected linker states between the two data sets. That is clearly not the case.

Seventhly, we are able to confirm claims made based on the negative stain 2D class averages by 3D cryoEM structures. In Figure 4A and B (original manuscript) we conclude that linker remodelling takes place at the interface between the linker stem and middle domains. We are able to validate this conclusion by the 3D cryoEM structures of the Rea1_{D2915A-R2976A-D3042A} salt-bridge mutant (Figure 6F and page 10, lines 8 -10; original manuscript).

Eightly, we are also able to confirm interactions we see in our negative stain 2D classes by complementary approaches. As we have pointed out above, one of the predictions of linker state 8 is the interaction between the linker top2/3 domain and the MIDAS domain. We are able to confirm this interaction by crosslinking mass spectrometry, immunoprecipitation and in-vitro pull-downs. If linker

state 8 would just be an artefact with no relation to the real biochemistry of Rea1 we would not have been able to confirm this interaction.

6. The relationship between this “remodeling” in state 1 and the AAA-engagement or nucleotide states is not supported. There is no evidence that anything in the manuscript is nucleotide-dependent or informs about nucleotide-dependent remodeling. Thus, this seems to me like a collection of snapshots of a mobile molecule, where we have no idea if they represent any functional states.

We are surprised that the reviewer criticises that the relationship between linker remodelling state 1 and the nucleotide state is not supported. We have never mentioned in the manuscript that state 1 is nucleotide dependent. On the contrary, at multiple instances in the manuscript (page 5, line 32; page 7, lines 19 – 23; page 12, lines 2 - 5; all original manuscript) as well as in several main text figures (Figures 1E, 4 and 8, original manuscript), we clearly point out that linker states 1 – 5 are part of the intrinsic conformationally flexibility of Rea1 and completely nucleotide independent. In light of the fact that this aspect was mentioned repeatedly in the manuscript, it is frankly speaking disturbing that the reviewer nevertheless states the complete opposite here.

The reviewer also questions the functional relevance of the alternative linker remodelling states, including state 1. This is likewise surprising and also confusing to us, since the reviewer just indirectly acknowledged the functional relevance of state 1 in point 5 (“Also, the functionality of this structure must be demonstrated not just using the salt bridge mutant.....”). Concerning the functional relevance of the other linker remodelling states, we refer to reviewer to our reply to point 2. The Rea1 mutant data we show in figures 6 and 7 (original manuscript) clearly demonstrate that mutants, which are no longer able to sample these linker remodelling states, lead with various functional defects. This correlation clearly supports the idea that the alternative Rea1 linker remodelling states are of functional importance.

7. The section on force-generation is completely overblown and lacks any controls. If they want to say anything about force, they need to show that this occurs with ATP, but not AMPPNP, or ADP. The difference between AMPPNP and ATP_γS must be also demonstrated in ATPase, remodeling and force assays.

It is not true that the microtubule gliding assays we used to demonstrate Rea1 force production did not have any controls. We refer the reviewer to page 22 line 19 (original manuscript), where we clearly state that we did not observe microtubule gliding events under APO conditions. At the request of the reviewer, we have now included additional control experiments, including an AMPPNP control, which also showed no microtubule movements (Figure 5E, revised manuscript). The difference between AMPPNP and ATP_γS in ATPase assays (our reply to point 2) as well as negative stain EM assays (Figure 2, original manuscript) has been already demonstrated. In our opinion it is not necessary to carry out the microtubule gliding assays in the presence of ATP_γS. The sole aim of these assays is to demonstrate that remodelling of the linker with respect to the AAA+ ring is able to produce mechanical force in an ATP dependent way. Whether or not in addition to ATP also ATP_γS is able to drive such force production is an irrelevant question in this context.

8. The IP does not show anything. Not even the expression of the constructs (input) is the same, the “correct” thing is only observed as a very minor species in lane 2. And there is no reason to assume that the minor bands in the pulldown are the TOP2/3 and not other crossreacting species. This needs to be either totally redone, or removed.

As we point out in the legend of supplementary figure 10 (original manuscript), the differences in the input lanes are due to the fact that all constructs are prone to degradation of various extent. Nevertheless, we detect an interaction between top2 and MIDAS fragments (lane 2). In contrast to the opinion of the reviewer, we are confident that these bands do represent top2 fragments, because they are not detected in lanes 3 and 4. Since the lysates originated from the identical yeast strain (only the transformed plasmids differed), we have the same amount of background proteins in all experiments. If what we detected in lane 2 was a crossreacting species from the protein background, we should have detected it also in 3 and 4, which is clearly not the case. Furthermore, we have now also independently confirmed the top2/3-MIDAS domain interaction by an in-vitro pulldown assay (Figure 5B, revised manuscript).

9. Interactions the authors suggest exist need to be tested with mutagenesis, as the resolution of all the structures, even the best ones is at best moderate.

*This comment indicates a misunderstanding of the reviewer in our opinion. We do not suggest any intra-molecular interaction based on the presented 3D cryoEM structures (Figures 3A and 6F, original manuscript). We use the *Rea1*_{ΔAAA2H2α} ATPγS cryoEM structure (Figure 3A, original manuscript) to generate 2D projections that allow us to do the domain assignment in our negative stain 2D class averages (Figure 3C, original manuscript). We use the cryoEM structures of the ATP salt bridge mutant (Figure 6F, original manuscript) to confirm our claim from the negative stain EM 2D class averages that linker remodelling occurs at the interface between the linker stem and middle domains (page 10, lines 8 -10, original manuscript). It is simply the property of these structures that they differ at the linker stem/middle domain interface (Figure 6F, original manuscript). We would like to emphasise again that in all these cases we do not claim that interactions between amino-acid residues stabilize something. We cannot test what we have never claimed.*

10. The only thing in the paper that might be a result are the findings with the salt bridge mutant. But, breaking something is very easy....can this be rescued if the salt bridges are reversed ie. Positive to negative and vice versa....importantly, I am also not convinced that the “remodeling” is actually critical, as it is observed without any nucleotide. That part needs to be better established. It just seems like wishful thinking.

The whole point of the salt-bridge mutant was to demonstrate the importance of the linker middle domain and to provide evidence for the functional relevance of the observed linker remodelling states. The disruption of this highly-conserved network impairs the rotation of the linker so that we do not detect the alternative linker remodelling states anymore (Figure 6B, original manuscript). As a consequence, we observe various functional defects (Figure 6E and 7, original manuscript). These results prove the functional relevance of the conserved salt-bridge network within the linker middle domain and suggest that the alternative linker remodelling states we report in this manuscript are of functional importance.

The additional mutagenesis experiments the reviewer suggests would address a completely different aspect of the salt-bridge mutant by targeting the question to which extent the conserved salt-bridge

network actually controls the rotational movement of the Rea1 linker. If reversing the electrostatics within the network does indeed rescue the rotational movement of the linker, it would prove that this rotation is solely controlled within the salt-bridge network. If such a reversal does not rescue the rotation, additional structural elements outside the salt-bridge network must be involved. Although we did not include this aspect in the manuscript, we agree that it is potentially interesting.

However, it would not add anything new to the aspects for why we included the salt-bridge mutant in this manuscript in the first place (see above). The question whether or not a charge reversal of the salt-bridge network will rescue the rotational movement is irrelevant for our statements concerning the general functional relevance of the salt-bridge network and the alternative linker states. It is also not relevant for any other conclusion in the manuscript. Thus, we feel that such experiments should be included in follow up work, preferably in combination with high-resolution structures highlighting the molecular basis for linker rotation.

With respect to the salt-bridge mutant the reviewer also criticises that linker remodelling might not be “critical, as it is observed without any nucleotide”. We would like to point out that this is again a very surprising and plainly wrong statement. We investigated linker remodelling of the salt-bridge mutant always in the presence of ATP (Figures 6B and F, original manuscript). In fact, we never investigated any other nucleotide state, so we cannot draw any conclusion about which linker remodelling states occur without nucleotide in case of the salt-bridge mutant. We were never interested in this question, the main reason for why we included the salt-bridge mutant in the manuscript is explained above. Nowhere in the manuscript we write that linker remodelling in case of the salt-bridge mutant occurs without nucleotide.

11. The significance of the crosslinking data is unclear. How many total crosslinks are there? Where are all the other crosslinks? Are they all consistent with the structures? They should be mapped in ways that highlight their abundance. Crosslinking is notoriously non-specific.

As we already pointed out in detail in our reply to point 2, the crosslinking mass spectrometry data validates our negative stain data of linker remodelling state 8 by detecting a linker top2/3 domain – MIDAS domain interaction. Furthermore, the data implicates a highly conserved MIDAS domain loop region in the interaction, which is essential for Rea1 mediated assembly factor removal. We feel that these aspects clearly highlight the significance of the crosslinking mass spectrometry data.

The number of total crosslinks and their corresponding sites are included in the data files that will be submitted to the mass spectrometry repository. We also have mentioned now the number of crosslinks for all data sets in the legend for supplementary figure 12 of the revised manuscript. For the available cryoEM structure of the straight linker conformation, we have mapped the crosslinks consistent with this structure for all available data sets (supplementary figure 12C, original manuscript). We do not see why it should be necessary to highlight the abundance of crosslinks. In the manuscript we do not refer to any crosslink except the ones between the linker top2 and the MIDAS domain. In our opinion it is more important to validate the crosslinks that are relevant for the manuscript with the appropriate control experiments, which we have already done (page 8, lines 10 – 11, original manuscript). To strengthen this point, we now include also an APO control in addition to the AMPPNP control of the original manuscript. In both cases, we do not detect the crosslinks between the linker top2 and the MIDAS domains, suggesting they specifically occur in the presence of ATP_γS (page 8, line 30-32, revised manuscript). These control experiments also address the crosslinking specificity issue raised by the reviewer.

12. The cryo-EM is lacking any documentation. How did the particles segregate? What fraction of the dataset was used?

It is simply not true that there is no documentation concerning the cryoEM in the manuscript. We provide FSC curves, local resolution maps as well as images showing the fit between structural models and corresponding cryoEM maps in supplementary figures 4 and 12 (original manuscript). We also describe how the cryoEM grid preparation, data collection and image processing were done (page 18, lines 13 – 34; page 19, lines 1 – 12; original manuscript). We also state the total number of particles in the data sets and the number of particles that were used to calculate the cryoEM maps shown in the manuscript (page 19, lines 5 – 9, original manuscript). This info would allow the attentive reader to calculate the fraction of particles used.

To summarize, as far as I can tell there are negative stain images, but too much made of percentages that are unlikely to be statistically significant and not bolstered by enough data. Negative stain is also really low resolution and not suitable for nature communications. There are a few moderate resolution structures whose significance is unclear, which suggest a remodeling of the linker in a nucleotide-independent manner. The specific remodeling needs to be confirmed using mutagenesis. Such mutants could support their significance in addition to the specific suggested structures. The entire matter of the nucleotide-dependent engagement with the AAA-ring is not supported by the data. Overall, I cannot see how this work is a significant advance.

Reviewer #4 (Remarks to the Author):

Busselez et al. study the ribosome maturation factor Rea1 which catalyzes the removal of assembly factors from large ribosomal subunit precursors to promote their export from the nucleus to the cytosol. The authors demonstrate that Rea1 rotates and swings towards the AAA+ ring following a complex remodeling scheme involving nucleotide independent as well as nucleotide dependent steps. They thus conclude, (i) that ATP-hydrolysis is required to engage the linker with the AAA+ ring and ultimately with the AAA+ ring docked MIDAS domain and (ii) that the interaction between the linker top and the MIDAS domain allows force transmission for assembly factor removal.

To probe if the described linker remodeling is able to produce mechanical force, the authors carried out microtubule gliding assays. They modified Rea1 Δ AAA2H2 α + Δ 4168-4907 by fusing a GFP (allowing anchorage to the cover slip surface) to AAA5 and a spytag (with dynein microtubule binding domain) into the linker top2 domain. After the application of fluorescently labelled microtubules, they screened for microtubule gliding events. The authors observed occasional events of slow, directed microtubule gliding over several minutes (Figure 5C, Movies S3-S5) suggesting that the remodeling of the linker top with respect to the AAA+ ring is able to produce mechanical force. They reason that the mechanical force produced by linker remodeling might be directly applied to the AAA+ ring-docked MIDAS domain via the linker top2/top3 domains to remove assembly factors from pre60S particles.

While I am not an expert to judge the quality of the overall results of the manuscript (which I nevertheless find to be carried out on a very high level), I was asked by the editor to provide a brief technical report, assessing the reported TIRF microscopy based microtubule gliding assay experiments.

In general, I find it a great idea to design and perform these experiments. Also, I tend to believe that what the authors see is directional motility. However, to solidify their statement, the authors should convince the readers that the reported events are indeed directional translocation and not the result of random diffusive motion. After all, currently only three rare events are shown, out of fields of view where most of the microtubules appear to be statically attached to the surface or are genuinely diffusing. The authors could, for example, perform a mean-square displacement (MSD) analysis proving that MSD vs. time cannot be fit by a linear function alone - but rather needs to contain a quadratic function.

Moreover, it should be added to the manuscript: (i) the number of total gliding events, along with how many statically attached and purely diffusing microtubules have been observed, (ii) the gliding velocities, (iii) scale bars in Figure 5C, and (iv) some discussion on the magnitude of the "forces". The forces needed to move microtubules as observed (short filaments with extremely low velocity on top of some random motion) are expected to be extremely low. Would these forces be sufficient to fulfill the ascribed function of assembly-factor removal?

The observed microtubule gliding events vary greatly in velocity, which inflates the error bars in the mean-squared displacement analysis as requested by the reviewer. We discussed this issue with a biophysicist in our institute and he suggested to provide an alternative MSD analysis that is independent of the actual microtubule gliding velocity, but still allows to distinguish between random diffusion and directed movement. He suggested to analyse the anomalous coefficient α , which is equal to 1 for a random diffusion event and greater than 1 in case of directed movement. The anomalous coefficient α can be obtain by logarithmic plotting of the mean-squared displacement MSD vs. time, fitting the data to a linear function and determine the slope of the linear function, which corresponds to α (see material and methods, page 27, lines 3 – 12, revised manuscript). We have done this analysis for 20 microtubule gliding events (Figure 5F and G, revised manuscript). The average α is 1.49 ± 0.20 , which is comparable to the α of the motor protein dynein known to drive directed microtubule gliding.

In addition to the statistical analysis requested by the reviewer, we have also carried out more control experiments to strengthen the point that the observed microtubule gliding events are not caused by random diffusion. In figure 5E of the revised manuscript, we now provide a statistical analysis of microtubule behaviour in the absence of the dynein microtubule binding domain (-MTBD, +ATP) and in the presence of the non-hydrolysable ATP analogue AMPPNP (+MTBD, +AMPPNP) in addition to the APO control (+MTBD, -ATP) already reported in the original manuscript. In all these cases, we do not observe microtubule gliding events suggesting that the observed events depend on the presence of the dynein MTBD as well as ATP and are not driven by random diffusion.

We have added the requested scale bars in Figure 5 of the revised manuscript.

In the revised manuscript, we provide a brief discussion about the magnitude of the force (page 13, lines 27 – 32). In our opinion, we cannot draw any conclusions on the force produced by individual Rea1 molecules since we used an ensemble assay to investigate microtubule gliding. We also know from our negative stain EM analysis that in the presence of ATP only a minority of Rea1 molecules

sample the conformational states associated with ATP hydrolysis that would allow force production. So, even if individual force production might be strong, collectively it would appear as weak.

REVIEWERS' COMMENTS

Reviewer #1 (Remarks to the Author):

The authors have put in a lot of effort to rework the manuscript both at the experimental and the written text levels.

Specifically, they have quantified the different conformational states of the Rea1 mutant and importantly outlined the caveats of their analyses in the Discussion. The positive control for the microtubule gliding assays also strengthens their data and model they propose regarding the Rea1 linker domain.

Altogether, they have addressed my concerns in a clear and satisfactory manner.

Reviewer #2 (Remarks to the Author):

The authors addressed all of my concerns very carefully and have improved the manuscript.

I was also asked by the editorial team to comment on the authors' response to reviewer 3. Below, I will address each point raised by Reviewer 3 individually:

1. The reviewer is concerned about the low frequency of ATP hydrolysis-driven linker states and questions the statistical relevance of the states with respect to the size of the data sets. The authors correctly point out that they already have included control experiments with large datasets. In the revised manuscript they add another control by analysing 390.000 negatively stained particles of the Rea1 Δ AAA2H2a in the presence of AMPPMP. In my opinion, the size of the control datasets is sufficient to support the authors' conclusion. The high signal-to-noise ratio in negative stain images enables much more sensitive detection of even small substates, so that the authors' careful and extensive analysis seems valid.

2. The reviewer is concerned about the significance of the low abundance states and questions the experimental set-up. The comment that AMPPMP is a slowly hydrolysable ATP analogue is simply not supported. To my knowledge AMPPNP cannot be hydrolysed by AAA+ proteins. The authors support this fact by providing a comparison between Rea1 hydrolysis rates for AMPPNP and ATP γ S, showing that AMPPNP is non-hydrolysable. Given the examination of so many different nucleotide states, I agree with the authors that they did the best they could to rule out that low abundance states arise from artefacts.

3. I don't understand the criticism in point 3 either. The authors did the best they could to make sense of it.
4. In the revised manuscript only the salt-bridge mutant is discussed and all requested functional tests are provided.
5. I agree with all 8 statements of the authors of why the alternative linker remodeling states cannot be the result of negative stain artefacts. The authors furthermore provide a side-by-side comparison of the negative stain and cryo-EM structures of the ATPgS bound state in response to the reviewers comments.
6. This point of reviewer 3 is unjustified, as all relevant information was already presented in the original submission.
7. The reviewer requests additional control experiments for the microtubule gliding assay, which the authors provide and which, in my opinion, show that the linker remodeling leads to a mechanical force (ATP dependent). The use of ATPgS in this assay most likely produces very slow movements, which might not be possible to record. The different effects of ATP, ATPgS and AMPPMP have already been shown in ATPase assays and in the structural experiments.
8. Due to the weak signal, the reviewer fears that the immunoprecipitation between TOP2/3 and the Midas domain is being overinterpreted. The authors provide an in vitro binding assay between the Midas domain and the linker TOP2/3 domain that demonstrates binding between the two fragments but not to GST. This further supports their original statement.
9. The reviewer requests a mutational study to confirm claimed interactions. However, due to the lack of high resolution information this endeavour would require a random mutagenesis analysis on a 5000 amino acid molecule. Of course it would be great to have a high resolution structure to deduce crucial side chain interactions that support the observed interaction between the Midas domain and the TOP2/3 domain. Only then it would make sense to target specific aa in functional tests. In the absence of such structures, the authors nevertheless have provided significant insight into the nucleotide dependent mechanics of Rea1. They used various methods to validate their low resolution structural data and filtered out some important observations from this data. All data are presented in a clear and careful way.
10. I agree with the authors that a charge reversal study for the salt bridge residues is beyond the scope of this manuscript.
11. The reviewer seems to doubt the validity of cross linking experiments in general. In my opinion they can provide powerful conclusions in combination with low resolution structural data. In response to the reviewers comments the authors include another control experiment of cross linked APO complexes further strengthening their observations.
12. The reviewer requests further information on the image analysis process of the cryo EM data presented. I understand that the reviewer prefers cryo EM data processing being displayed in a flow chart rather than text format. Given that the data processing was done without special sorting, symmetries or advanced segmentation methods I think it is justified to do without a flow chart in this case. All processing steps can be retraced using the information provided.

I have no further concerns and recommend acceptance of the manuscript.

Reviewer #4 (Remarks to the Author):

The authors have addressed my comments satisfactorily and I have no further comments.

REVIEWERS' COMMENTS

Reviewer #1 (Remarks to the Author):

The authors have put in a lot of effort to rework the manuscript both at the experimental and the written text levels.

Specifically, they have quantified the different conformational states of the Rea1 mutant and importantly outlined the caveats of their analyses in the Discussion. The positive control for the microtubule gliding assays also strengthens their data and model they propose regarding the Rea1 linker domain.

Altogether, they have addressed my concerns in a clear and satisfactory manner.

Reviewer #2 (Remarks to the Author):

The authors addressed all of my concerns very carefully and have improved the manuscript.

I was also asked by the editorial team to comment on the authors' response to reviewer 3. Below, I will address each point raised by Reviewer 3 individually:

1. The reviewer is concerned about the low frequency of ATP hydrolysis-driven linker states and questions the statistical relevance of the states with respect to the size of the data sets. The authors correctly point out that they already have included control experiments with large datasets. In the revised manuscript they add another control by analysing 390.000 negatively stained particles of the Rea1deltaAAA2H2a in the presence of AMPPMP. In my opinion, the size of the control datasets is sufficient to support the authors' conclusion. The high signal-to-noise ratio in negative stain images enables much more sensitive detection of even small substates, so that the authors' careful and extensive analysis seems valid.
2. The reviewer is concerned about the significance of the low abundance states and questions the experimental set-up. The comment that AMPPMP is a slowly hydrolysable ATP analogue is simply not supported. To my knowledge AMPPNP cannot be hydrolysed by AAA+ proteins. The authors support this fact by providing a comparison between Rea1 hydrolysis rates for AMPPNP and ATPgammaS, showing that AMPPNP is non-hydrolysable. Given the examination of so many different nucleotide states, I agree with the authors that they did the best they could to rule out that low abundance states arise from artefacts.
3. I don't understand the criticism in point 3 either. The authors did the best they could to make sense of it.
4. In the revised manuscript only the salt-bridge mutant is discussed and all requested functional tests are provided.
5. I agree with all 8 statements of the authors of why the alternative linker remodeling states cannot be the result of negative stain artefacts. The authors furthermore provide a side-by-side comparison of the negative stain and cryo-EM structures of the ATPgS bound state in response to the reviewers comments.
6. This point of reviewer 3 is unjustified, as all relevant information was already presented in the original submission.
7. The reviewer requests additional control experiments for the microtubule gliding assay, which the authors provide and which, in my opinion, show that the linker remodeling leads to a mechanical force (ATP dependent). The use of ATPgS in this assay most likely produces very slow movements,

which might not be possible to record. The different effects of ATP, ATPgS and AMPPMP have already been shown in ATPase assays and in the structural experiments.

8. Due to the weak signal, the reviewer fears that the immunoprecipitation between TOP2/3 and the Midas domain is being overinterpreted. The authors provide an in vitro binding assay between the Midas domain and the linker TOP2/3 domain that demonstrates binding between the two fragments but not to GST. This further supports their original statement.

9. The reviewer requests a mutational study to confirm claimed interactions. However, due to the lack of high resolution information this endeavour would require a random mutagenesis analysis on a 5000 amino acid molecule. Of course it would be great to have a high resolution structure to deduce crucial side chain interactions that support the observed interaction between the Midas domain and the TOP2/3 domain. Only then it would make sense to target specific aa in functional tests. In the absence of such structures, the authors nevertheless have provided significant insight into the nucleotide dependent mechanics of Rea1. They used various methods to validate their low resolution structural data and filtered out some important observations from this data. All data are presented in a clear and careful way.

10. I agree with the authors that a charge reversal study for the salt bridge residues is beyond the scope of this manuscript.

11. The reviewer seems to doubt the validity of cross linking experiments in general. In my opinion they can provide powerful conclusions in combination with low resolution structural data. In response to the reviewers comments the authors include another control experiment of cross linked APO complexes further strengthening their observations.

12. The reviewer requests further information on the image analysis process of the cryo EM data presented. I understand that the reviewer prefers cryo EM data processing being displayed in a flow chart rather than text format. Given that the data processing was done without special sorting, symmetries or advanced segmentation methods I think it is justified to do without a flow chart in this case. All processing steps can be retraced using the information provided.

I have no further concerns and recommend acceptance of the manuscript.

Reviewer #4 (Remarks to the Author):

The authors have addressed my comments satisfactorily and I have no further comments

We are pleased to hear that all reviewers have no further requests.